# Bootstrapping Language Models with DPO Implicit Rewards

**Changyu Chen**[*1], **Zichen Liu**[*23], **Chao Du**[†2], **Tianyu Pang**[2], **Qian Liu**[2],
**Arunesh Sinha**[†4], **Pradeep Varakantham**[†1], **Min Lin**[2]
[1]Singapore Management University    [2]Sea AI Lab, Singapore
[3]National University of Singapore    [4]Rutgers University
{chency,liuzc,duchao,tianyupang,liuqian,linmin}@sea.com;
arunesh.sinha@rutgers.edu; pradeepv@smu.edu.sg

## Abstract

Human alignment in large language models (LLMs) is an active area of research. A recent groundbreaking work, direct preference optimization (DPO), has greatly simplified the process from past work in reinforcement learning from human feedback (RLHF) by bypassing the reward learning stage in RLHF. DPO, after training, provides an implicit reward model. In this work, we make a novel observation that this implicit reward model can by itself be used in a bootstrapping fashion to further align the LLM. Our approach is to use the rewards from a current LLM to construct a preference dataset, which is then used in subsequent DPO rounds. We incorporate two refinements to further improve our approach: 1) length-regularized reward shaping to make the preference dataset length-unbiased; 2) experience replay to enhance the quality of the preference dataset. Our approach, named self-alignment with **D**PO **I**mpli**C**it r**E**wards (DICE), shows great improvements in alignment. It achieves an increase of more than $8\%$ in length-controlled win rate on AlpacaEval 2 for all the different base models that we tried, without relying on external feedback. Our code is available at https://github.com/sail-sg/dice.

## 1 Introduction

Direct preference optimization (DPO) (Rafailov et al., 2024b) presents a compelling alternative to reinforcement learning from human feedback (RLHF) (Stiennon et al., 2020) in large language models (LLMs). By circumventing the complexity of learning a reward model from given human preferences, DPO is simpler to implement and train compared to the RLHF approaches. Importantly, DPO, once trained, implicitly specifies a reward model. Mathematically, the reward for a response $\mathbf{y}$ to the prompt $\mathbf{x}$ can be expressed in terms of the optimal policy $\pi^\star$ and the reference policy $\pi_{\text{ref}}$:

$$r^\star(\mathbf{x}, \mathbf{y}) = \beta \log \frac{\pi^\star(\mathbf{y}|\mathbf{x})}{\pi_{\text{ref}}(\mathbf{y}|\mathbf{x})} + \beta \log Z(\mathbf{x}),$$

for parameter $\beta$ and normalizing constant $Z$. Further, an *implicit reward* $r(\mathbf{x}, \mathbf{y}) = \beta \cdot [\log \pi^\star(\mathbf{y}|\mathbf{x}) - \log \pi_{\text{ref}}(\mathbf{y}|\mathbf{x})]$ is defined in DPO where the normalizing constant term can be ignored as it will be canceled out in the DPO objective, which only involves the difference of the rewards for the same prompt. In this work, we explore whether the above readily available implicit reward model after DPO training provides an opportunity to further improve the language model.

This paper answers the research question in the affirmative, by using the above implicit rewards in a *bootstrapping* fashion to further improve the LLM alignment with human preferences. Specifically, our approach follows the iterative DPO framework (Tran et al., 2023), where implicit rewards serve as the preference signals, as illustrated in Figure 1. We start with a model that has been through one round of DPO using human preference data, referred to as a DPO-tuned model. We then use the implicit rewards induced by DPO itself to rank outputs from the current LLM, thereby, yielding a new

---

[*]Equal contribution. The project was done during Changyu Chen's internship at Sea AI Lab.
[†]Correspondence to Chao Du, Arunesh Sinha, and Pradeep Varakantham.

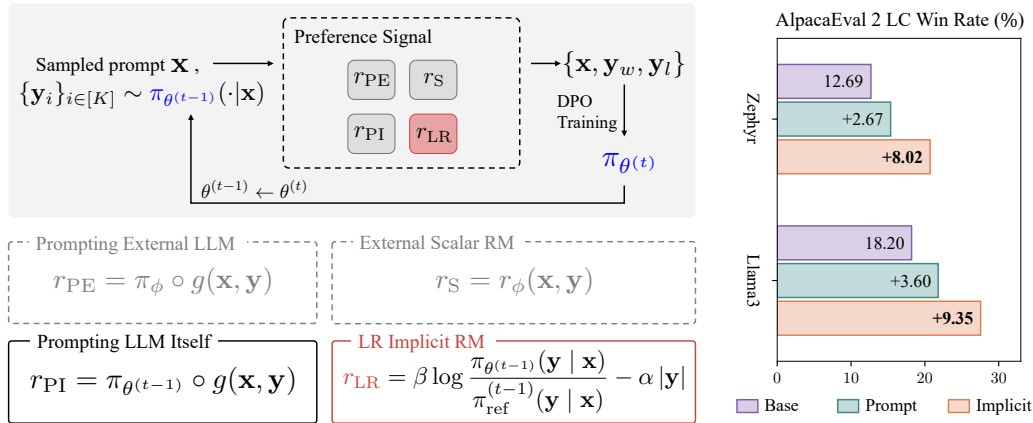

Figure 1: **(Left)** Iterative DPO (Tran et al., 2023) with various preference signals: under the iterative DPO framework, the policy model is iteratively trained on a newly generated preference dataset. This dataset can be constructed using various preference signals. A common source is a scalar reward model (RM) (Ouyang et al., 2022), denoted as $r_\phi$. Alternatively, the dataset can be created by prompting a LLM to judge the responses. This LLM can either be an external model $\pi_\phi$ or the policy model itself from the previous iteration $\pi_{\theta^{(t-1)}}$. In this context, $\mathbf{x}$ and $\mathbf{y}$ are processed through a LLM-as-a-judge template $g(\cdot, \cdot)$ (Yuan et al., 2024). We propose utilizing the length-regularized (LR) implicit rewards introduced in Section 3.1, where $\pi_{\theta^{(t-1)}}$ and $\pi_{\text{ref}}^{(t-1)}$ represent the policy model and reference model from the previous DPO iteration, respectively. Our experiment excludes approaches requiring external models, such as $r_\phi$ and $\pi_\phi$, as they are beyond this work's scope. **(Right)** Our method, which leverages implicit rewards, further improves DPO-tuned models by a large margin, resulting in superior performance compared to the prompting counterpart.

preference dataset cheaply. We run DPO again with this newly generated preference dataset to obtain an updated LLM and then repeat the process. However, the above approach still needs further refinement to address practical issues. One is the known issue of length exploitation (Park et al., 2024) where LLMs generate long responses when the same content could be provided more succinctly. Another issue is that the implicit reward model is an approximate proxy for human preferences, hence relying on it strongly can result in corruption of the initial knowledge inbuilt into the LLM.

Inspired by (Park et al., 2024), we address the length exploitation issue by length-regularized reward shaping, which discourages long responses from being preferred. Notably different from Park et al. (2024), who incorporate response length into the loss function, we directly search for a length-unbiased dataset, avoiding the expensive hyper-parameter tuning. To fix the overreliance on implicit rewards, we leverage insights from continual learning (Rolnick et al., 2019) and replay high quality human preference data that was used in the first round of DPO (the round before bootstrapping began). Our method, named self-alignment with **D**PO Impli**C**it r**E**wards (DICE), significantly improves LLM alignment quality with different base models. On AlpacaEval 2, we achieve $8.02\%$ length-controlled (LC) win rate improvement with the Zephyr-based model and $9.35\%$ improvement with the Llama3-based model.

To summarize, our main contributions are as follows:

- We propose to utilize the implicit reward model readily available in a DPO-tuned LLM. The implicit reward model enables us to evaluate the responses generated by the current policy and construct a preference dataset for future rounds of DPO without any external feedback;

- We propose to apply two techniques together with our above proposed approach, length-regularized reward shaping and experience replay;

- Empirical results show that our approach DICE enables significant (more than $8\%$ LC win rate increase on AlpacaEval 2) improvement in alignment with different base models; thus, we believe that the core idea of using DPO Implicit Reward in DICE is a general purpose approach that can improve alignment for any single DPO-tuned base model.

## 2 PRELIMINARIES

We provide a brief review of the standard RLHF (Stiennon et al., 2020; Ouyang et al., 2022) and DPO algorithms (Rafailov et al., 2024b). Through the review, we demonstrate the implicit reward model induced by DPO, which will be used in our work.

In preference tuning, the preference data typically takes the form of pairwise preferences. Each prompt $\mathbf{x}$ is paired with two possible responses, $\mathbf{y}_1$ and $\mathbf{y}_2$. The human annotator (Christiano et al., 2017) or AI annotator (Lee et al., 2023) provides the preference feedback $o(\mathbf{y}_1 \succ \mathbf{y}_2|\mathbf{x}) \in \{0, 1\}$, indicating whether $\mathbf{y}_1$ is preferred over $\mathbf{y}_2$. The preferred response is denoted as $\mathbf{y}_w$, while the other is denoted as $\mathbf{y}_l$. A common assumption is that the ground-truth human preferences follow the Bradley-Terry model (Bradley & Terry, 1952). Based on this assumption, we can train a parameterized reward model $r_\phi(\mathbf{x}, \mathbf{y})$ using maximum likelihood:

$$\mathcal{L}_R\left(r_\phi, \mathcal{D}\right) = -\mathbb{E}_{(\mathbf{x}, \mathbf{y}_w, \mathbf{y}_l) \sim \mathcal{D}}\left[\log \sigma\left(r_\phi\left(\mathbf{x}, \mathbf{y}_w\right) - r_\phi\left(\mathbf{x}, \mathbf{y}_l\right)\right)\right], \tag{1}$$

where $\sigma$ is the logistic function.

### 2.1 REINFORCEMENT LEARNING FROM HUMAN FEEDBACK

The standard RLHF algorithm treats the LLM as a policy and optimizes the policy using the reward model $r_\phi$. The optimization objective is represented by the following equation:

$$\max_{\pi_\theta} \mathbb{E}_{\mathbf{x} \sim \mathcal{D}, \mathbf{y} \sim \pi_\theta(\mathbf{y}|\mathbf{x})}[r_\phi(\mathbf{x}, \mathbf{y})] - \beta \cdot \mathbb{D}_{\text{KL}}\left[\pi_\theta(\mathbf{y}|\mathbf{x})\|\pi_{\text{ref}}(\mathbf{y}|\mathbf{x})\right], \tag{2}$$

where $\pi_{\text{ref}}(\mathbf{y}|\mathbf{x})$ denotes a reference distribution, and $\beta$ is a hyper-parameter. This objective balances the maximization of the reward $r_\phi(\mathbf{x}, \mathbf{y})$ and divergence from the fixed reference distribution. The divergence term, given by the KL divergence (i.e., $\mathbb{D}_{\text{KL}}\left[\pi_\theta(\mathbf{y}|\mathbf{x})\|\pi_{\text{ref}}(\mathbf{y}|\mathbf{x})\right]$) acts as a regularizer to prevent the policy $\pi_\theta$ from drifting too far away from the initial distribution $\pi_{\text{ref}}(\mathbf{y}|\mathbf{x})$. This objective is then optimized using a general-purpose RL algorithm, such as PPO (Schulman et al., 2017).

### 2.2 DIRECT PREFERENCE OPTIMIZATION

DPO (Rafailov et al., 2024b) starts with the same objective as Eq. (2) but derives an analytical closed-form relation between the reward and the resulting optimal policy. This relation can be used to reparameterize the ground truth reward in terms of the corresponding optimal policy. This reparameterized formulation can be substituted back into the reward optimization objective in Eq. (1), enabling direct training of the optimal model on the feedback data using the following objective:

$$\mathcal{L}_{\text{DPO}}\left(\pi_\theta; \pi_{\text{ref}}\right) = -\mathbb{E}_{(\mathbf{x}, \mathbf{y}_w, \mathbf{y}_l) \sim \mathcal{D}}\left[\log \sigma\left(\beta \log \frac{\pi_\theta\left(\mathbf{y}_w|\mathbf{x}\right)}{\pi_{\text{ref}}\left(\mathbf{y}_w|\mathbf{x}\right)} - \beta \log \frac{\pi_\theta\left(\mathbf{y}_l|\mathbf{x}\right)}{\pi_{\text{ref}}\left(\mathbf{y}_l|\mathbf{x}\right)}\right)\right]. \tag{3}$$

In this context, the parameter $\beta$ is the same as in Eq. (2), balancing the expected reward and divergence from the initial model. The DPO objective is particularly advantageous as it facilitates the recovery of the optimal model through a standard classification loss, without the need for on-policy sampling or extensive hyper-parameter tuning. Observe that Eq. (3) resembles the reward modeling objective in Eq. (1) under the parameterization

$$r(\mathbf{x}, \mathbf{y}) = \beta \log \frac{\pi_\theta(\mathbf{y}|\mathbf{x})}{\pi_{\text{ref}}(\mathbf{y}|\mathbf{x})}. \tag{4}$$

This reward function is commonly referred to as an "implicit reward" (Rafailov et al., 2024a; Zhong et al., 2024). Theorem 1 in Rafailov et al. (2024b) demonstrates that this parameterization of a reward model is indeed valid without loss of generality. If we substitute this form of $r_\theta(\mathbf{x}, \mathbf{y})$ into the RL objective in Eq. (2), we can derive the optimal solution in a closed form, which is $\pi_\theta$. Consequently, once DPO optimization is completed, we obtain an "implicit reward model" as defined by Eq. (4).

## 3  BOOTSTRAPPING WITH DPO IMPLICIT REWARDS

DPO is an attractive alternative to RLHF as it largely simplifies the implementation and training process of language model alignment. However, recent evidences (Guo et al., 2024; Tran et al., 2023) have shown that continuing DPO training on a fixed offline dataset results in inferior policy, while the policy can be further improved if one can collect new responses generated by the updated policy and provide preference labels to perform another round of DPO training. This can be understood as being closer to the on-policy sampling, which is generally preferred in preference fine-tuning (Tajwar et al., 2024; Tang et al., 2024).

In this work, we employ such iterative DPO preference tuning framework, where we start from a base language model (a base policy) $\pi_{\theta^{(0)}}$ that is DPO-tuned from an initial reference model $\pi_{\text{ref}}^{(0)}$, commonly an SFT model. In each round $t \in \{1, 2, \dots\}$, we sample $K$ on-policy responses from the latest policy $\pi_{\theta^{(t-1)}}(\cdot \mid \mathbf{x})$ given a prompt $\mathbf{x}$. We then label the response with the highest and the lowest implicit rewards as winning and losing responses respectively, thus constructing a new preference dataset $\mathcal{D}_t$. We further fine-tune the policy with DPO's objective (Eq. (3)) to obtain the updated policy $\pi_{\theta^{(t)}}$ with reference model $\pi_{\text{ref}}^{(t)} = \pi_{\theta^{(t-1)}}$. This process is repeated to iteratively improve the language model. Notably, by fine-tuning a DPO-tuned LM on the its own constructed dataset, we essentially align it without relying on any external preference feedback (e.g., RLHF or RLAIF), hence in a *bootstrapping* fashion. We thus refer to our method as iterative self-alignment (bootstrapping) with **D**PO **I**mpli**C**it r**E**wards (DICE).

We next introduce two ingredients that are proven critical in the self-alignment process. In Section 3.1, we present Length-Regularized Implicit Rewards, which augment the vanilla implicit rewards with a length-regularized reward shaping, to judge the on-policy sampled responses for constructing the preference dataset. Furthermore, to mitigate the potential catastrophic forgetting in the continual fine-tuning, we propose experience replay (Section 3.2) that mixes the generated data with the offline data for better performance.

### 3.1  LENGTH-REGULARIZED IMPLICIT REWARDS

It is a known issue in the literature that preference tuning may introduce length bias (or length exploitation) (Park et al., 2024) , which is likely caused by the fact that the preference labels collected from human annotators favor more verbose responses. This problem is further compounded by the iterative self-alignment scheme such as the one in Yuan et al. (2024), because the generated responses that are long and preferred will be reinforced in the next round of DPO, leading the language model to generate increasingly longer responses.

Inevitably, the vanilla DPO implicit rewards as in Eq. (4) would also exhibit length bias when generating preference dataset. In Figure 2, we show the distribution of the difference in string length of the winning and losing responses. We can see from the top figure that vanilla implicit rewards yield a skewed distribution (in green), with an average length difference 1031. In stark contrast, the length difference of a high quality preference dataset is almost normally distributed (as in the bottom figure). This observation motivates us to debias the distribution induced by vanilla implicit rewards so as to mitigate the length exploitation. We resort to reward shaping (Sutton & Barto, 2018) for this purpose. In particular, we introduce a length-regularized (LR) reward shaping term in the implicit

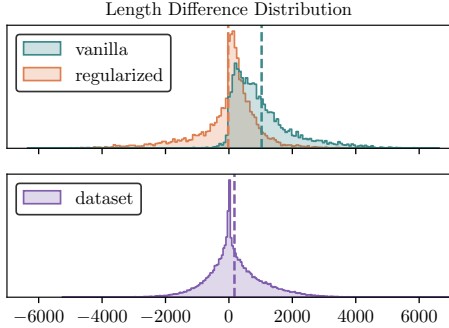

Figure 2: Distribution of the length difference between the winning and losing examples ($|\mathbf{y}_w| - |\mathbf{y}_l|$). **(Top)** Distribution of the first round on-policy generated dataset. With LR reward shaping defined by Eq. (5), the length bias is mitigated and the length difference becomes more evenly distributed. The average length difference decreases from 1031 to $-21$ by setting $\alpha$=0.023. **(Bottom)** Distribution of the high quality UltraFeedback preference dataset (Cui et al., 2023) is almost unbiased.

---

**Algorithm 1** Bootstrapping with DPO Implicit Rewards (DICE)

---

1: **Input:** prompt set $\mathcal{X}$ extracted from $\mathcal{D}_{\text{offline}}$, initial DPO-tuned policy $\pi_{\theta^{(0)}}$, initial reference policy $\pi_{\text{ref}} = \pi_{\theta^{(-1)}}$, number of generated samples $K$, regularization weight $\beta$, experience replay weight $\gamma \in (0, 1)$

2: **for** $t = 1, 2, \ldots$ **do**

3:     Generate responses by sampling $\mathbf{x} \sim \mathcal{X}$ and $\mathbf{y}_{1:K} \sim \pi_{\theta^{(t-1)}}$;

4:     Create preference dataset $\mathcal{D}(\alpha^{\star})$ by optimizing Eq. (6);
    // $\mathcal{D}(\alpha)$ is constructed by taking the best and worst $\mathbf{y}$ based on $r_{\text{LR}}(\mathbf{x}, \mathbf{y}_k; \alpha)$, $k \in [K]$;
    // Evaluate $r_{\text{LR}}(\mathbf{x}, \mathbf{y}_k; \alpha)$ given $\pi_{\theta^{(t-1)}}$ and $\pi_{\theta^{(t-2)}}$ as the target policy and reference policy

5:     Create the mixed dataset via experience replay $\mathcal{D}_t = \{(\mathbf{x}^i, \mathbf{y}_w^i, \mathbf{y}_l^i)\}_{i \in [N]}$;
    // Sample $(\mathbf{x}, \mathbf{y}_w, \mathbf{y}_l) \sim p_{\mathcal{D}_t}$, $p_{\mathcal{D}_t} = (1 - \gamma)p_{\mathcal{D}(\alpha^{\star})} + \gamma p_{\mathcal{D}_{\text{offline}}}$

6:     Optimize $\pi_\theta$ according to DPO loss, Eq. (3):

$$\theta^{(t)} \leftarrow \arg\min_{\theta} -\mathbb{E}_{(\mathbf{x}, \mathbf{y}_w, \mathbf{y}_l) \sim \mathcal{D}_t} \left[ \log \sigma \left( \beta \log \frac{\pi_\theta(\mathbf{y}_w | \mathbf{x})}{\pi_{\theta^{(t-1)}}(\mathbf{y}_w | \mathbf{x})} - \beta \log \frac{\pi_\theta(\mathbf{y}_l | \mathbf{x})}{\pi_{\theta^{(t-1)}}(\mathbf{y}_l | \mathbf{x})} \right) \right];$$

7: **end for**

---

reward that penalizes the length of the response to obtain the shaped reward:

$$r_{\text{LR}}(\mathbf{x}, \mathbf{y}; \alpha) = \beta \log \frac{\pi_\theta(\mathbf{y} | \mathbf{x})}{\pi_{\text{ref}}(\mathbf{y} | \mathbf{x})} - \alpha |\mathbf{y}|, \tag{5}$$

where $\alpha$ controls the penalty strength and $|\mathbf{y}|$ is the string length of the response. Based on the shaped rewards, we can construct many versions of the preference dataset $\mathcal{D}(\alpha)$, following the principle that the response with the higest $r_{\text{LR}}(\mathbf{x}, \mathbf{y}_i; \alpha)$ is labeled as $\mathbf{y}_w$ and the one with the lowest reward is labeled as $\mathbf{y}_l$. To find the most suitable $\alpha$ such that $\mathcal{D}(\alpha)$ is (approximately) unbiased, we optimize $\alpha$ with the objective to minimize the average absolute difference in response length:

$$\alpha^{\star} = \arg\min_{\alpha} \left| \mathbb{E}_{(\mathbf{y}_w, \mathbf{y}_l) \sim \mathcal{D}(\alpha)} (|\mathbf{y}_w| - |\mathbf{y}_l|) \right|. \tag{6}$$

We can employ any black-box optimizer to solve this non-differentiable objective function. In this work we find a simple random search suffices, and its solution effectively transforms the dataset into a more evenly distributed one (shown in the orange curve in the top of Figure 2). We will output $\mathcal{D}(\alpha^{\star})$ for the next round of DPO training. Details of the optimization can be found in Appendix C.

Importantly, despite the resemblance of Eq. (5) to Park et al. (2024) where they incorporate the token length as a regularizer in the training objective, our reward shaping is conducted during the dataset construction stage, thereby avoiding the need for expensive hyper-parameter tuning.

## 3.2 EXPERIENCE REPLAY

Though DICE enables us to learn from the response of the current policy, we know that the implicit reward model from the DPO training is not a perfect proxy for human preferences. Solely relying on the implicit reward model may result in forgetting the knowledge inbuilt in the initial policy at the first DPO stage. Inspired by the technique of experience replay (Rolnick et al., 2019) in continual learning for preventing catastrophic forgetting and making a good balance between old and new data, we propose to use a mixture of the generated data and the offline preference dataset. While offline preference data is considered to be high-quality, it is off-policy samples; the generated data is closer to the on-policy samples, but the imperfect implicit reward model may introduce noise in labeling the preference. Combining the two can make for a good balance. This idea is also related to deep Q-learning with demonstrations (Hester et al., 2018), which demonstrates that incorporating supervised learning from an offline dataset can accelerate online reinforcement learning.

Our complete algorithm is summarized in Algorithm 1. During each iteration $t$, we generate $K$ responses $\mathbf{y}_1, \mathbf{y}_2, \ldots, \mathbf{y}_K$ from the policy $\pi_{\theta^{(t-1)}}$ for each prompt $\mathbf{x}$ (Line 3). The preference dataset $\mathcal{D}(\alpha)$ is constructed based on the LR rewards $r_{\text{LR}}$, where the response with the highest LR reward is labeled $\mathbf{y}_w$ and the one with the lowest reward is labeled $\mathbf{y}_l$. In this process, $\pi_{\theta^{(t-1)}}$ serves as the target policy, and $\pi_{\theta^{(t-2)}}$ serves as the reference policy. For $t = 1$, we denote the reference policy in the implicit reward model as $\pi_{\theta^{(-1)}}$, corresponding to the reference policy used in the initial DPO training. At Line 4, we construct the debiased dataset $\mathcal{D}(\alpha^{\star})$ by minimizing the average absolute

difference in response length, referring to Eq. (6). Subsequently, a mixed dataset $\mathcal{D}_t$ is created by sampling $\gamma$ proportion of data from $\mathcal{D}_{\text{offline}}$ and $(1 - \gamma)$ proportion of data from $\mathcal{D}(\alpha^\star)$ (Line 5). Finally, DPO training is conducted on $\mathcal{D}_t$ using $\pi_{\theta^{(t-1)}}$ as both the initial policy and the reference policy, resulting in the updated policy $\pi_{\theta^{(t)}}$ (Line 6).

# 4 EXPERIMENTS

This section empirically investigates DICE. Our findings highlight several key points: (1) DICE significantly improves the model performance on the widely used leaderboard AlpacaEval 2 (Li et al., 2023b), increasing length-controlled win rate by more than $8\%$ for all the different base models; (2) our best model bootstrapped from a 8B base model (`Llama-3-8B-DPO`) achieves a better performance than Gemini Pro 1.0 (Team et al., 2023) without any extra human annotations (or external reward model) other than the preference dataset that is used in the initial DPO training of base model; (3) the two proposed techniques in Sections 3.1 and 3.2 are shown to be critical for DICE; (4) DICE demonstrates competitive performance relative to scalar reward models trained exclusively on the same seed data.

## 4.1 EXPERIMENT SETUP

**Base Models and Datasets.** We adopt `Llama-3-8B-DPO`[1] and `zephyr-7B-beta`[2] as our base models. Both models are trained following the pipeline of Zephyr (Tunstall et al., 2023) on the UltraFeedback (Cui et al., 2023) dataset. `Llama-3-8B-DPO` is trained based on meta-llama/Meta-Llama-3-8B , developed by Meng et al. (2024). `zephyr-7B-beta` is fine-tuned based on mistralai/Mistral-7B-v0.1 . We randomly sample a subset of around 10k preference pairs from UltraFeedback as the offline dataset $\mathcal{D}_{\text{offline}}$ for our fine-tuning experiments. Our experiments aim to show how much the language model can improve from a DPO-tuned model and a subset of the preference dataset that was used to conduct the initial DPO training.

**Response Generation and Dataset Construction.** At the start of each round, we sample responses from the current policy, with temperature $T = 0.9$, $p = 1.0$ for the Llama3 setting and $T = 0.7$, $p = 0.9$ for the Zephyr setting. We sample with different random seeds to get $K = 16$ diverse responses for each prompt. We then reward each prompt-response pair by the implicit reward model (Eq. (4)) and incorporate length-regularized reward shaping (Eq. (5)) to get the debiased dataset $\mathcal{D}(\alpha^\star)$ with the optimal regularization strength $\alpha^\star$. The final dataset is a mixture of $\mathcal{D}(\alpha^\star)$ and $\mathcal{D}_{\text{offline}}$.

**Training Details.** All experiments are conducted on 8 Nvidia A100 GPUs. For DICE, we trained two rounds in total. In each round, we train the model for 300 steps on a preference dataset with 9.6k preference pairs (either a solely generated dataset, or a mixture of the generated dataset and the offline preference dataset). The global training batch size is set to 32 and the learning rate is 5e-7 with a constant schedule and a warm-up of 50 steps. We hypertuned $\beta \in \{0.01, 0.1\}$ based on the model performance on AlpacaEval 2 for each method and model separately. For our approach, we additionally hypertuned the experience replay ratio $\gamma$ using cross-validation to ensure fair assessment. Further details on hyperparameter tuning can be found in Appendix F.

**Baselines.** We evaluate the following baseline methods applicable to the setting of this paper:

- Offline DPO: continue conducting DPO training with the offline preference dataset.

- Offline DPO w/ new ref: similar to offline DPO but we assign the trained policy after each round as the new reference model for the next round. This corresponds to $\gamma = 1$.

- DPO with prompted rewards: similar to self-rewarding LM (Yuan et al., 2024), where they prompt the LLM itself as a preference judge to construct new preference pairs and iteratively fine-tune the LLM with the DPO algorithm. In their case, the judge capability is learned by supervised fine-tuning on an evaluation fine-tuning dataset. We exploit our base model to perform LLM-as-a-Judge directly as it can follow the judge instructions well. In the experiment, we call it *LLM-as-a-Judge*. The LLM-as-a-Judge prompt template can be found in Appendix E.

---

[1] https://huggingface.co/princeton-nlp/Llama-3-Base-8B-SFT-DPO
[2] https://huggingface.co/HuggingFaceH4/zephyr-7b-beta

Table 1: Results of AlpacaEval 2 and Arena-Hard across two base models. LC and WR denote length-controlled and raw win rate in percentage (%) respectively.

| Method | zephyr-7B-beta | | | Llama-3-8B-DPO | | |
| | AlpacaEval 2 | | Arena-Hard | AlpacaEval 2 | | Arena-Hard |
| | LC | WR | LC | LC | WR | LC |
|---|---|---|---|---|---|---|
| Base | 12.69 | 10.71 | 10.17 | 18.20* | 15.50* | 22.32 |
| Offline DPO Iter 1 | 13.40 | 11.10 | 14.13 | 20.22 | 18.33 | 24.78 |
| Offline DPO Iter 2 | 4.96 | 5.47 | 1.89 | 21.04 | 19.21 | 24.04 |
| Offline DPO (w/ new ref) Iter 1 | 13.40 | 11.10 | 13.10 | 22.29 | 19.96 | 22.2 |
| Offline DPO (w/ new ref) Iter 2 | 4.58 | 5.27 | 5.15 | 22.50 | 20.18 | 23.55 |
| LLM-as-a-Judge Iter 1 | 15.36 | 17.81 | 13.96 | 20.30 | 21.31 | 26.46 |
| LLM-as-a-Judge Iter 2 | 14.14 | 17.89 | 13.47 | 21.80 | 22.42 | 27.99 |
| DICE Iter 1 | 19.03 | 17.67 | 18.15 | 25.08 | 25.77 | 33.52 |
| DICE Iter 2 | **20.71** | **20.16** | **18.45** | **27.55** | **30.99** | **39.13** |

* We note that the results of Llama-3-8B-DPO base are obtained from Meng et al. (2024).

**Evaluation.** We evaluate our method by AlpacaEval 2 (Li et al., 2023b) and Arena-Hard (Li et al., 2024). Both are LLM-based automatic evaluation benchmarks and have been widely adopted by the community. AlpacaEval 2 employs AlpacaFarm (Dubois et al., 2023) as its prompts set composed of general human instructions. The model responses and the reference responses generated by GPT-4-Turbo are fed into a GPT-4-Turbo-based annotator to be judged. We follow the standard approach and report both the win rate (WR) and the Length-Controlled win rate (LC) (Dubois et al., 2024) over the reference responses. Arena-Hard is a recently released benchmark, incorporating 500 well-defined technical problem-solving queries. Due to the expensive evaluation cost of Arena-Hard, we follow the official guidance and use a strong open-source model mistralai/Mistral-Large-Instruct-2407 (123B)[3] as the judge model.

## 4.2 MAIN RESULTS

**DICE Effectively Improves a DPO-tuned Model.** With only a DPO-tuned model and a preference dataset that was used to train this model, one can choose to further improve the current policy via Offline DPO, or construct a new preference dataset via LLM-as-a-Judge. In Table 1, we compare the performance of the model fine-tuned by DICE in two rounds with the base model and other baselines. It shows all methods can improve the LC win rate on AlpacaEval 2 over the base model while DICE leads to the most significant enhancement in both Zephyr and Llama3 settings, increasing by **8.02**% and **9.35**% respectively. We found that the LLM-as-a-Judge leads to good performance in the Zephyr setting while it has minor improvement in the Llama3 setting. We hypothesize this may be caused by the coarse rewards which are not able to provide effective preference signals

Table 2: AlpacaEval 2 leaderboard results.

| Model | LC | WR |
|---|---|---|
| GPT-4 0613 | 30.18 | 15.76 |
| Mistral Medium | 28.61 | 21.86 |
| Claude 2 | 28.15 | 17.19 |
| DICE-Llama3 8B Iter 2 | 27.55 | 30.99 |
| Snorkel (Mistral-PairRM-DPO) | 26.39 | 30.22 |
| Gemini Pro | 24.38 | 18.18 |
| Mixtral 8×7B v0.1 | 23.69 | 18.26 |
| Llama 3 8B Instruct | 22.92 | 22.57 |
| GPT-3.5 Turbo 0613 | 22.35 | 14.10 |
| Tulu 2+DPO 70B | 21.24 | 15.98 |
| DICE-Zephyr 7B Iter 2 | 20.71 | 20.16 |
| GPT-3.5 Turbo 1106 | 19.30 | 9.18 |
| Llama-3-8B-DPO | 18.20 | 15.50 |
| Vicuna 33B v1.3 | 17.58 | 12.71 |
| zephyr-7B-beta | 12.69 | 10.71 |

when responses are of high quality (the prompt template requires LLM judge to provide a discrete score from 0 to 5, referring to Appendix E). We note that training on a fixed offline dataset for multiple rounds leads to even worse performance than the base model, possibly due to the increased data staleness and overfitting.

---

[3] https://huggingface.co/mistralai/Mistral-Large-Instruct-2407

Table 3: AlpacaEval 2 results showing DPO rewards are compatible with other DAP algorithms.

| Method | Offline | | DICE | |
|--------|---------|------|---------|------|
|        | LC | WR | LC | WR |
| DPO | 13.40 | 11.10 | 19.03 | 17.67 |
| IPO | 14.83 | 16.16 | 18.51 | 19.49 |
| KTO | 13.92 | 10.65 | 14.88 | 12.16 |
| Hinge | 13.51 | 12.45 | 15.92 | 15.57 |

Table 4: AlpacaEval 2 results showing effects of the Length Regularized Weight.

| $\gamma$ | $\alpha$ | LC | WR | Avg. Len. |
|------|--------------------------|-------|-------|-----------|
| 0.0 | 0.000 | 13.32 | 15.37 | 2570 |
|     | 0.023 ($\alpha^\star$) | 18.88 | 19.31 | 2109 |
|     | 0.046 ($2\alpha^\star$) | 14.40 | 9.08 | 876 |
| 0.5 | 0.000 | 15.92 | 19.08 | 2600 |
|     | 0.023 ($\alpha^\star$) | 19.03 | 17.67 | 1848 |
|     | 0.046 ($2\alpha^\star$) | 14.91 | 10.80 | 1185 |

**Comparison with the models on the leaderboard.** Compared with the models on the public leaderboard shown in Table 2, DICE-Llama3 8B performs better than the official instruct version of Llama3 by a non-trivial margin, $4.63\%$. Regarding the closed-source models, it achieves better performance than Gemini Pro with only 8B parameters and does not require any in-house data or external reward model.

### 4.3 DICE IS COMPATIBLE WITH OTHER DIRECT ALIGNMENT FROM PREFERENCE ALGORITHMS

Though DICE works best with DPO as it makes the iterative training possible (because the implicit reward model for the next round can be naturally derived using the updated policy), we would like to check if the dataset generated by DICE can also improve the base model with other Direct Alignment from Preference (DAP) algorithms. In the Zephyr setting, we tune the policy model using DICE-generated dataset (at the first round) and the offline dataset with KTO (Ethayarajh et al., 2024), IPO (Azar et al., 2024), and Hinge loss proposed in Zhao et al. (2023). The training follows the protocol described in Section 4.1. The results in Table 3 show that all DAP algorithms benefit from the newly generated data by the current policy and DPO implicit reward model with LR reward shaping, demonstrating LC win rates higher than their offline counterparts. Notably, DICE shows the greatest improvement for DPO.

### 4.4 COMPARING DICE WITH INTERNAL REWARD MODEL

Our experiment setup assumes access to an offline preference dataset and a DPO-tuned language model. To further improve the language model, one may choose to train a new scalar reward model and then conduct iterative DPO. As this scalar reward model is trained solely on the offline preference dataset without utilizing the external resources, we call it the internal reward model (IntlRM). In this section, we compare the IntlRM with implicit reward model to evaluate their performance in our self-alignment setting.

**Base Models and Datasets.** We utilize `mistral-7b-sft-beta`[4] as the base model. We adopt the full UltraFeedback preference dataset (Cui et al., 2023), comprising 60k preference pairs.

**Training Details.** For the implicit reward, we use `zephyr-7B-beta` along with its reference model `mistral-7b-sft-beta`. The scalar reward model is trained using OpenRLHF (Hu et al., 2024) framework, adhering to the recommended training parameters[5]. This setup ensures that both the implicit reward model and IntlRM use the same base model (`mistral-7b-sft-beta`) and are trained on the same dataset (UltraFeedback).

**Evaluation.** We evaluate the reward models based on the alignment rate of the model with the preference labels provided by GPT-4o. The alignment rate is defined as $m/n$, where $n$ is the total number of prompt-response pairs and $m$ is the number of labels matching GPT-4o. As our goal is to provide preference labels for the model-generated responses to enable the subsequent DPO training, we sample 500 $(\mathbf{x}, \mathbf{y}_1, \mathbf{y}_2)$ tuples from the preference dataset during the first iteration of the DICE experiment under the Zephyr setting. Different reward models are then queried to provide the preference labels.

---

[4] https://huggingface.co/HuggingFaceH4/mistral-7b-sft-beta
[5] https://github.com/OpenRLHF/OpenRLHF/blob/main/examples/scripts/train_rm_llama.sh

Table 5: Alignment rate of the reward model with the preference labels provided by GPT-4o.

| Method | IntlRM | ERM-555k | DPO Implicit Reward |
|---|---|---|---|
| Alignment Rate | 0.624 | 0.656 | **0.698** |

In the experiment, we include an external reward model, ERM-555k[6], trained by OpenRLHF maintainers on 555k preference data, as a stronger baseline. In Table 5, we observe that the DPO implicit reward achieves higher accuracy than the IntlRM trained on an equivalent amount of preference data. Furthermore, it surpasses the performance of the ERM-555k trained on substantially more data. This implies that the implicit reward model offers advantages when evaluating its own generated data, though the scalar reward models in general excel on a wider range of tasks when the preference data is abundant, as demonstrated by ArmoRM-Llama3-8B-v0.1, which was trained on 1M preference data. Based on these findings, we argue that the implicit reward is a competitive option in self-alignment settings.

### 4.5 ABLATION STUDY

In this section, we investigate the effects of LR reward shaping and experience replay.

**Effects of LR reward shaping.** LR reward shaping (Eq. (5)) penalizes responses for being too verbose, and guides the construction of a debiased dataset with the optimal penalty strength $\alpha^\star$ found by optimizing Eq. (6). To validate the effectiveness of the propose LR reward shaping as well as the $\alpha$-searching procedure, we run experiments in the Zephyr setting with different mixture ratios ($\gamma = 0$ and $\gamma = 0.5$) and ablate three design choices: (1) no LR reward shaping ($\alpha = 0$); (2) LR reward shaping with penalty strength found by Eq. (6), i.e., $\alpha = \alpha^\star = 0.023$; (3) LR reward shaping with slightly larger penalty strength ($\alpha = 2\alpha^\star$). Results are presented in Table 4. For all values of the $\gamma$, we observe that $\alpha = 0.0$ does lead to serious length exploitation. So, even if the policy can get a high win rate, it will suffer in the LC win rate due to the length exploitation issue (responses with longer average length get a lower LC win rate), e.g., the low LC win rate with $\gamma = 0.5, \alpha = 0.0$. In contrast, a larger $\alpha$ seemingly mitigates the length exploitation issue even better, but it may adversely affect the response quality. Our proposed approach of finding $\alpha^\star$ using the objective of minimizing the absolute difference of response length does provide the best performance.

**Effects of experience replay.** The experience replay results in a new mixed dataset in which $\gamma$ fraction of the data is from the offline dataset and $(1 - \gamma)$ fraction of the data is from the generated dataset. E.g., we use data only from the generated dataset if $\gamma = 0.0$. We run experiments in the Zephyr setting with $\gamma \in \{0.0, 0.25, 0.5, 0.75, 1.0\}$ and conduct DICE in total of two rounds. From the results shown in Figure 3, we find $\gamma = 0.5$ provides the best performance. The results satisfy our expectations. With only offline preference data, the DPO optimizes the current policy with off-policy samples that are further away from its distribution. With only its own generated data, the model may keep reinforcing its current "belief", potentially leading to catastrophic forgetting. An intermediate value $\gamma = 0.5$ finds a good balance. Additional results in the Llama3 setting can be found in Appendix D.

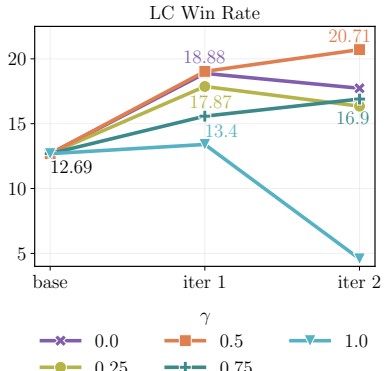

Figure 3: AlpacaEval 2 LC Win Rate across different experience replay ratios ($\gamma$) in the Zephyr setting. The highest LC Win Rate is reported via text.

## 5 LIMITATION AND FUTURE WORK

**Limitations.** One of the primary limitations of our work is the reliance on the DPO training prior to bootstrapping. If the implicit reward model is not well-trained, it can lead to a collapse of the training pipeline. This challenge is not unique to our approach; the classic RLHF pipeline is also

---

[6]https://huggingface.co/OpenRLHF/Llama-3-8b-rm-mixture

struggling when its reward model is not well-trained. Another limitation is the lack of continued improvement over many iterations. Similar to other research, such as the work by Yuan et al. (2024), which enhances the policy model without an external reward model, we did not observe continuous improvement in our model beyond three iterations. This issue highlights an open question within this field regarding the iterative enhancement of policy models.

**Future Work.** Future research could explore the rewarding capabilities of models trained using other DPO variants, such as KTO and IPO. Investigating whether these variants can offer general rewards similar to those provided by DPO-tuned policy would be valuable. Another promising direction involves developing methods that enable continuous improvement of the policy model over iterations without degradation. Additionally, investigating a theoretical understanding of the policy learned through self-bootstrapping could provide deeper insights into the mechanics of our approach and facilitate further advancements.

## 6    RELATED WORK

**Self-Improving Fine-Tuning.** Many studies explore fine-tuning language models without a large amount of human annotation (Huang et al., 2022; Li et al., 2023a; Sun et al., 2023; 2024; Yuan et al., 2024; Chen et al., 2024b; Zhang et al., 2025). Starting from an SFT model, Sun et al. (2023) collect the preference labels by prompting the SFT model itself to choose the preferred response based on principles, training a principle-driven reward model, and optimizing via PPO. Yuan et al. (2024) construct a preference dataset by their own supervised fine-tuned model trained on instruction following data and evaluation fine-tuning data, followed by DPO training. Different from these approaches, which align models by leveraging pretrained knowledge with minimal seed data, our work focuses on further improving a DPO-tuned model through bootstrapping with its implicit rewards.

**On-Policy Sampling in Preference Tuning.** DPO and its variants are widely used for their simplicity, but their offline nature often limits policy learning (Guo et al., 2024). Guo et al. (2024) show that offline DPO quickly overfits the preference dataset, while it performs much better and is more stable if online feedback can be provided to their on-policy samples[7]. Tajwar et al. (2024) further analyze the properties of different preference tuning methods, concluding that on-policy sampling enhances performance and efficiency. When pure on-policy sampling is infeasible, using data closer to on-policy samples, such as in iterative DPO with external rewards (Tran et al., 2023; Dong et al., 2024; Xiong et al., 2024; Ding et al., 2024; Pang et al., 2024; Muldrew et al., 2024) or internal rewards (Yuan et al., 2024; Furuta et al., 2025), can still be beneficial. Similarly, our approach enables us to train the policy model with the preference data which is closer to on-policy samples than the offline dataset without any external reward models. We hypothesize this is one of the main gain sources of our approach.

**DPO Implicit Rewards.** Rafailov et al. (2024a) recently studied DPO from a token-level MDP perspective, revealing that DPO-tuned models implicitly parameterize token-wise dense reward functions. Building on similar theoretical fundations, Zhong et al. (2024) pretrains a DPO model to serve as a standalone dense reward generator for PPO training. Unlike these works, we leverage the implicit reward model as an outcome reward model to bootstrap itself for self-alignment. Implicit reward models have also been used for selecting the pairwise data (Yang et al., 2024; Chen et al., 2024a) to improve the annotation efficiency in preference tuning.

## 7    CONCLUSION

In this paper, we introduce DICE, a novel approach that leverages the implicit reward model from DPO to further align LLMs with human preferences. Our method stands out in the current landscape of LLM alignment research, as it uses the implicit reward model to iteratively refine the policy model. Empirical results show that our approach DICE enables significant (more than $8\%$ LC win rate increase on AlpacaEval 2) improvement in alignment across different base models, without relying on any external feedback.

---

[7]On-policy samples: the data collected while following the current policy that is being optimized.

ETHICS STATEMENT

This paper presents work whose goal is to advance the field of Machine Learning. There are many potential societal consequences of our work, none of which we feel must be specifically highlighted here.

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

# A    BENEFITS OF ON-POLICY SAMPLING

In this section, we provide a theoretical analysis similar to Xie et al. (2024) to show that learning with on-policy samples can be more effective than utilizing an offline dataset. We make a generalized notation for the sampling policy at the $t$-th round as $\pi^{(t)}$, and compare the on-policy sampling ($\pi^{(t)} = \pi_{\theta^{(t-1)}}$) with sampling from an offline dataset ($\pi^{(t)} = \pi_\mu$[8]). For a prompt $\mathbf{x}$, we denote its optimal response[9] as $\mathbf{y}^\star$ and suboptimal ones as $\mathcal{S} = \{\mathbf{y}_i^-\}$. We also denote its preferences sampled from $\pi^{(t)}$ as $(\mathbf{y}_w^{(t)}, \mathbf{y}_l^{(t)})$. By Eq. (3) and the definition of the logistic function, we can rewrite the DPO loss at round $t$ for the sample $(\mathbf{x}, \mathbf{y}_w^{(t)}, \mathbf{y}_l^{(t)})$ as:

$$\mathcal{L}_{\mathrm{DPO}}^{(t)}(\pi_{\theta^{(t)}}; \pi_{\mathrm{ref}}^{(t)}) = -\log \frac{\pi_{\theta^{(t)}}\left(\mathbf{y}_w^{(t)} \mid \mathbf{x}\right)^\beta}{\pi_{\theta^{(t)}}\left(\mathbf{y}_w^{(t)} \mid \mathbf{x}\right)^\beta + \pi_{\theta^{(t)}}\left(\mathbf{y}_l^{(t)} \mid \mathbf{x}\right)^\beta R^\beta}, \tag{7}$$

where $R = \pi_{\mathrm{ref}}^{(t)}(\mathbf{y}_w^{(t)} \mid \mathbf{x})/\pi_{\mathrm{ref}}^{(t)}(\mathbf{y}_l^{(t)} \mid \mathbf{x})$ is a constant. Observe that Eq. (7) can be minimized to zero by just minimizing $\pi_{\theta^{(t)}}(\mathbf{y}_l^{(t)} \mid \mathbf{x})$ to be zero, without minimizing the likelihood of other suboptimal responses $\pi_{\theta^{(t)}}(\mathbf{y}^- \mid \mathbf{x})$ for $\mathbf{y}^- \neq \mathbf{y}_l^{(t)}$. After $t$ rounds of optimization, we are interested in the probability of outputting the optimal response:

$$\pi_{\theta^{(t)}}(\mathbf{y}^\star \mid \mathbf{x}) = 1 - \sum_{\mathbf{y}_i^- \in \mathcal{S}} \pi_{\theta^{(t)}}(\mathbf{y}_i^- \mid \mathbf{x}). \tag{8}$$

Eq. (8) allows us to reveal the deficiency of training on a fixed offline dataset. If there exists a suboptimal response $\mathbf{y}^- \in \mathcal{S}$ that lies in the high likelihood region of $\pi_{\theta^{(t)}}$, say $\pi_{\theta^{(t)}}(\mathbf{y}^- \mid \mathbf{x}) \geq p$ for some $p \in [0,1]$ that is close to 1, and $\mathbf{y}^-$ is never sampled from $\pi^{(t)} = \pi_\mu$ thus not optimized as $\mathbf{y}_l^{(t)}$ during all $t$ rounds, we have:

$$\pi_{\theta^{(t)}}(\mathbf{y}^\star \mid \mathbf{x}) \leq 1 - \pi_{\theta^{(t)}}(\mathbf{y}^- \mid \mathbf{x}) \leq 1 - p. \tag{9}$$

Since $\pi_\mu$ is zero except points that appear in $\mathcal{D}_{\mathrm{offline}}$, it is highly likely to find such a $\mathbf{y}^-$ not being sampled from $\pi_\mu$ (hence never optimized) and therefore $\pi_{\theta^{(t)}}(\mathbf{y}^\star \mid \mathbf{x})$ can be very low with a large $p$.

On the other hand, conducting on-policy sampling can alleviate the "never-sampled" issue and promote convergence to the optimal policy. This is because whenever $\pi_{\theta^{(t-1)}}(\mathbf{y}_i^- \mid \mathbf{x})$ is high, it is likely to sample $\mathbf{y}_i^-$ from $\pi^{(t)} = \pi_{\theta^{(t-1)}}$ and thus it can be optimized such that $\pi_{\theta^{(t)}}\left(\mathbf{y}_l^{(t)} = \mathbf{y}_i^- \mid \mathbf{x}\right) \approx 0$. In this way, the subtrahend of Eq. (8) is decreased per round, hence we can gradually improve the language model policy towards the optimal policy.

# B    EXPERIMENT BEYOND TWO ITERATIONS

We conduct DICE and LLM-as-a-judge baseline (the strongest baseline) for three iterations, evaluating them on AlpacaEval 2 benchmark (see Table 6).

From the table, we observe a drop in LC win rate at the third iteration for both our approach and the LLM-as-a-Judge baseline. This is a known challenge in the literature; both self-rewarding language model (Yuan et al., 2024) and SPPO (Wu et al., 2024) show successful improvement only up to three iterations, which corresponds to two iterations in our approach, as our method begins after one initial iteration of DPO training. Additionally, we use a fixed prompt set in every iteration, whereas other works employ different prompt sets across iterations. We hypothesize that using different prompt sets across iterations could help mitigate the performance drop observed in the third iteration.

---

[8] $\pi_\mu$ is the behavior policy that collects the offline dataset, which can be a mixture of existing LLMs or human experts.

[9] Optimality is measured by the underlying BT reward model, instead of the implicit rewards. In general there can be multiple optimal responses, but here we assume a single optimal response for simplicity.

Table 6: Results of AlpacaEval 2 across two base models over three iterations. LC and WR denote length-controlled and raw win rate in percentage (%) respectively.

| Method | zephyr-7B-beta AlpacaEval 2 | | Llama-3-8B-DPO AlpacaEval 2 | |
|---|---|---|---|---|
| | LC | WR | LC | WR |
| Base | 12.69 | 10.71 | 18.20* | 15.50* |
| LLM-as-a-Judge Iter 1 | 15.36 | 17.81 | 20.30 | 21.31 |
| LLM-as-a-Judge Iter 2 | 14.14 | 17.89 | 21.80 | 22.42 |
| LLM-as-a-Judge Iter 3 | 13.95 | 17.23 | 16.86 | 19.78 |
| DICE Iter 1 | 19.03 | 17.67 | 25.08 | 25.77 |
| DICE Iter 2 | **20.71** | **20.16** | **27.55** | **30.99** |
| DICE Iter 3 | 14.02 | 15.30 | 20.61 | 26.18 |

*We note that the results of Llama-3-8B-DPO base are obtained from Meng et al. (2024).

## C  OPTIMIZATION FOR LENGTH REGULARIZED REWARD SHAPING

We solve the objective in Eq. (6) using a simple Bayesian optimization toolkit based on Gaussian process[10]. The objective landscape with respect to $\alpha$ is depicted in Figure 4, where we compare the proposed implicit reward model with the LLM-as-a-Judge reward model. With our length-reguarlized implicit rewards, the optimizer is able to find the optimal solution quickly that debiases the length difference of the winning and losing responses. For LLM-as-a-Judge rewards, the optimal solution is obtained with $\alpha = 0$, hence we do not explicitly debias the dataset for all the experiments.

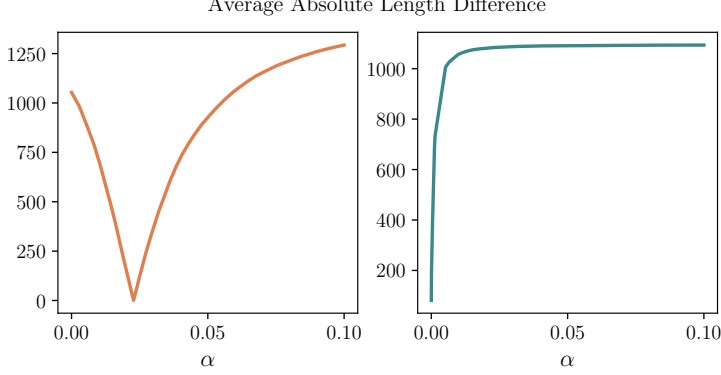

Figure 4: The objective landscape for (**Left**) our implicit reward model and (**Right**) the LLM-as-a-Judge reward model.

## D  EXTENDED ABLATION STUDY

**Effects of experience replay with Llama3 backbone model.** In the Llama3 setting, we also conduct a coarse sweeping for the experience ratio $\gamma$, and present the AlpacaEval 2 LC win rate in Figure 5 for two self-alignment rounds. We observe similar trends to those in the Zephyr setting, which further justify the effectiveness of the proposed experience replay: it helps to keep a balance between the more on-policy generated data and the curated offline data. The best identified value of the mixture ratio is $\gamma = 0.1$.

---

[10]https://scikit-optimize.github.io/stable/modules/generated/skopt.gp_minimize.html

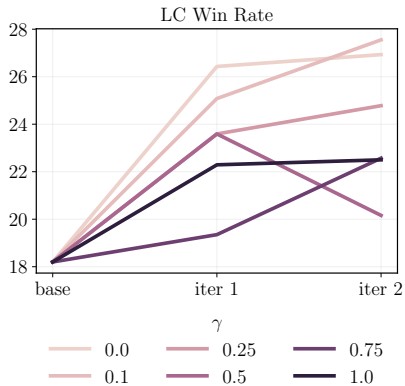

Figure 5: AlpacaEval 2 LC Win Rate across different experience replay ratio ($\gamma$) in the Llama3 setting.

**Comparison between Length-Regularized DPO and LR reward shaping.** The length-regularized DPO proposed by Park et al. (2024) incorporates a penalty term into the DPO objective to explicitly penalize length exploitation. In contrast, our approach mitigates this issue by applying reward shaping during preference dataset construction. Since both methods target the same problem, we compare their performance in the Zephyr setting with $\gamma = 0$. To clarify notation, we denote the length regularization coefficient in Park et al. (2024) as $\lambda$. Length-regularized DPO uses the generated dataset $\mathcal{D}(0)$ without reward shaping, and $\lambda$ is hypertuned from $\{0.02, 0.05\}$ as suggested in length-regularized DPO paper. In comparison, our method employs the dataset $\mathcal{D}(\alpha^\star)$. All other settings remain consistent across both approaches.

As shown in Table 7, we observe that our approach achieves better performance while effectively mitigating the length exploitation. In contrast, regularized DPO exhibits a quality-length trade-off, where larger regularization coefficients successfully control response length but lead to a decline in response quality.

Table 7: Results of AlpacaEval 2 comparing techniques for mitigating length exploitation. LC and WR denote length-controlled and raw win rate in percentage (%) respectively. Len denotes the average string length of the model response.

| Method | $\lambda$ | AlpacaEval 2 | | |
|---|---|---|---|---|
| | | LC | WR | Len |
| No mitigation | - | 13.32 | 15.37 | 2570 |
| Regularized DPO | 0.02 | 16.30 | 19.16 | 2629 |
| | 0.05 | 16.03 | 16.14 | 2030 |
| Ours | - | **18.88** | 19.31 | 2109 |

**PPO with Implicit Reward.** To assess whether implicit rewards can be effectively utilized in reward model-based methods such as PPO (Schulman et al., 2017), we conduct experiments using PPO with an implicit reward model. Specifically, the `zephyr-7B-beta` model and its SFT model are used to construct the implicit reward model. Additionally, `zephyr-7B-beta` serves as the base model to be optimized. We adopt the implementation in OpenRLHF (Hu et al., 2024), adhering to the recommended training parameters. Evaluation results on AlpacaEval 2 are presented in Table 8.

The results indicate a slight improvement in length-controlled win rate for PPO, although the improvement is less pronounced compared to DPO. While PPO's performance can potentially be enhanced with additional hyperparameter tuning, such efforts would require substantial resources and time. We leave this for future work.

## E    PROMPT USED BY LLM-AS-A-JUDGE

We provide the prompt used by LLM-as-a-judge in Figure 6.

Table 8: Results of AlpacaEval 2 for Zephyr backbone. LC and WR denote length-controlled and raw win rate in percentage (%) respectively.

| Method | AlpacaEval 2 | |
|--------|------|------|
| | LC | WR |
| Base | 12.69 | 10.71 |
| PPO | 13.32 | 8.55 |
| DPO | 19.03 | 17.67 |

---

**LLM-as-a-Judge Prompt Template**

Review the user's question and the corresponding response using the additive 5-point scoring system described below. Points are accumulated based on the satisfaction of each criterion:

- Add 1 point if the response is relevant and provides some information related to the user's inquiry, even if it is incomplete or contains some irrelevant content.
- Add another point if the response addresses a substantial portion of the user's question, but does not completely resolve the query or provide a direct answer.
- Award a third point if the response answers the basic elements of the user's question in a useful way, regardless of whether it seems to have been written by an AI Assistant or if it has elements typically found in blogs or search results.
- Grant a fourth point if the response is clearly written from an AI Assistant's perspective, addressing the user's question directly and comprehensively, and is well-organized and helpful, even if there is slight room for improvement in clarity, conciseness or focus.
- Bestow a fifth point for a response that is impeccably tailored to the user's question by an AI Assistant, without extraneous information, reflecting expert knowledge, and demonstrating a high-quality, engaging, and insightful answer.

User: {instruction}
Response: {response}

After examining the user's instruction and the response, provide brief step-by-step justifications and conclude with a score between 0 to 5. You must follow the format below:

Evaluation: <evaluation>
Score: <score>

---

Figure 6: We follow the prompt template used by Yuan et al. (2024) to use LLMs to judge model responses and construct paired dataset for further preference tuning.

## F  HYPERPARAMETER TUNING

In our experiments, the hyperparameters, including $\gamma$ and $\beta$, were selected based on the model performance on AlpacaEval 2 (AE2). These parameters were tuned for each method and model separately (with $\gamma$ specific to our approach) but remained fixed across iterations. Specifically, $\beta$ was tuned during the first iteration from the set $\{0.01, 0.1\}$, while $\gamma$ was tuned using the ranges $\{0.0, 0.25, 0.50, 0.75, 1.0\}$ for the Zephyr backbone and $\{0.0, 0.10, 0.25, 0.50, 0.75, 1.0\}$ for the Llama-3 backbone. The best hyperparameter setting identified in AlpacaEval 2 was directly applied in Arena-Hard (AH) for all methods and models without any further tuning.

Compared to baseline approaches, our approach has the additional hyperparameter $\gamma$ which may offer potential advantages during evaluation when hypertuning is performed on AE2. To address this concern, we treat the two benchmarks (AE2 and AH) as proxies for validation and test sets to ensure fair comparisons. Specifically, we validate on AE2 while testing on AH, and then reverse the roles of the two benchmarks. Below we show the results in Tables 9 and 10:

Table 9: Model performance on AlpacaEval 2 (AE2) and Arena-Hard (AH) across different $\gamma$ for the Zephyr Backbone. LC denotes the length-controlled win rate. CI denotes confidence interval.

| $\gamma$ | LC. AE2 | LC.AH: Score (95% CI) |
|------|---------|-----------------------|
| 0.00 | 18.88 | 17.6 (-1.4, 1.7) |
| 0.25 | 17.87 | 16.1 (-1.2, 1.4) |
| **0.50** | **20.71** | **18.4** (-1.3, 1.6) |
| 0.75 | 16.90 | 14.9 (-1.6, 1.4) |
| 1.00 | 13.40 | 13.0 (-1.3, 1.3) |

Table 10: Model performance on AlpacaEval 2 (AE2) and Arena-Hard (AH) across different $\gamma$ for the Llama-3 Backbone. LC denotes the length-controlled win rate. CI denotes confidence interval.

| $\gamma$ | LC. AE2 | LC.AH: Score (95% CI) |
|------|---------|-----------------------|
| 0.00 | 26.93 | 39.1 (-2.3, 2.5) |
| **0.10** | **27.55** | **39.1** (-2.0, 2.4) |
| 0.25 | 24.78 | 33.3 (-2.1, 1.8) |
| 0.50 | 23.59 | 32.2 (-2.3, 2.2) |
| 0.75 | 22.57 | 27.4 (-2.0, 1.8) |
| 1.00 | 22.50 | 24.5 (-1.7, 1.8) |

The two benchmarks and two backbone models constitute four validation-test instances. We observe that the $\gamma$ selected from the validation set/benchmark consistently achieves the best performance on the test set across all instances, demonstrating the optimal value of $\gamma$ is robust to the choice of the validation set/benchmark. Meanwhile, we do observe that the optimal $\gamma$ for different backbone models is varied, which is expected, as different backbones produce responses of varying quality. Higher-quality generated data reduces the need for replayed experience, aligning with our observations for the Llama-3 backbone.

