# OpenReview forum: "Bootstrapping Language Models with DPO Implicit Rewards"
_ICLR.cc/2025/Conference — ICLR 2025 Poster_

### Official Review · Reviewer_3Zep · 2024-11-01

**Soundness:** 3
**Presentation:** 3
**Contribution:** 3
**Rating:** 8
**Confidence:** 4

**Summary:**

The authors propose to use the implicit reward model learned via DPO to create a preference dataset and further improve the policy learned by an extra DPO step on the newly created preference dataset. They propose two additional techniques: 1) length regularization to prevent the known issue of response length increasing through DPO and 2) incorporating human preference data among the preference data collected with the final policy from the previous iteration and annotated by the implicit RM from the previous iteration.

**Strengths:**

Writing is mostly good and clear (but see problems below).

Length bias is clearly illustrated and mitigated by the length penalty (though not a technical contribution as discussed below).

Table 4 presents some good results: one can have short responses scoring rather high (higher than the $\alpha=0$ baseline).

Results are good on AlpacaEval and Arena Hard.

EDIT: I raised my score significantly after authors addressed major concerns of mine. I thank authors for the diligent work on making the submission better and fostering proper credit assignment.

**Weaknesses:**

A key weakness in the current state is that authors should connect their approach for length bias mitigation to existing approaches: it seems to me that this is the exact same approach as in Park et al, 2024 (which is cited but credit is not sufficiently assigned as of now). While I understand that the use of the length penalty is new, it currently reads as if authors came up with the penalty from Park et al, which is unfair to them. I make a non-exhaustive list of the places where credit / claims must be clarified:
* “We address the length exploitation issue by length-regularized reward shaping, which discourages long responses from being preferred”
    * I understand this sentence as the length regularization being an actual contribution, which is not the case. Please fix.
* “We propose utilizing the length regularized (LR) implicit rewards introduced in Section 4.2”
    * these are the same as in Park et al, so again this sentence is misleading
* “We propose to apply two techniques together with our above proposed approach, length regularized reward shaping and experience replay”
    * same: applying an existing technique (which was developed in a very close context) is not a technical contribution here
* “Next, in Section 4.2, we introduce the proposed Length-Regularized Implicit Rewards, which augment the vanilla implicit rewards with a length-regularized reward shaping, to judge the on-policy sampled responses for constructing the preference dataset”
    * same comment
* “Importantly, despite the resemblance of Eq. (8) to Park et al. (2024) where they incorporate the token length as a regularizer in the training objective, our reward shaping is conducted during the dataset construction stage, thereby avoiding the need for expensive hyper-parameter tuning” -> this needs to be explicited before! and does not prevent from toning down the novelty claims on the length bias mitigation

In the same vein, why not using the regularized DPO from Park et al as a baseline?? If the length bias mitigation proposed works better, this would make for a good contribution of the paper, but then this should be backed by explicit comparisons.

I also find that the authors did not get to the bottom of the lack of improvement past iteration 2 yet:
* “we did not observe continuous improvement in our model beyond three iterations” -> why is this not presented in graphs? that is important data that is currently (afaict) not in the paper
* Unclear why there is no improvement after K iterations (K > 2), is it because the policy becomes too deterministic? any root cause analysis would be valuable here

Mathematical notations and statements
* I do not understand why using $\hat{r}$ instead of $r^*$ (this is the optimal reward!) l. 34
* “For a prompt x, we denote its optimal response as y^⋆”
    * in general there can be multiple optimal responses (as there can be multiple optimal policies), please adapt the statement
    * but here authors mean preferred response? this really needs clarification

Writing
* “In this way, the subtrahend of Eq. (6)”
* “To validate the effectiveness of the propose LR reward”

Clarity
* Section 4.1 as a whole is really confusing in the current state.
* citation missing for RLHF (Christiano et al) l. 28
* when mentioning the length exploitation issue, authors could quickly explain why the models exploit length (which is because longer responses are correlated with higher probability of being preferred by human or machine-based preferences)
* no citation for Gemini Pro! also the version used should be stated (1.0 vs 1.5 vs 1.5 002)
* “Offline DPO w/ new ref: similar to offline DPO but we assign the current policy as the new reference model, while we use a fixed reference model in Offline DPO. This corresponds to γ = 1” -> misleading as my understanding is that the reference is fixed (it is just the end policy of the first round of DPO)

Experiments / ablations missing
* What about using all the K samples with a Plackett-Luce model? Does this bring any benefit?
* Replaying human preference data seems similar to KL regularization to the previous policy -> would be interesting to compare
* Baseline missing: offline AI feedback annotations instead of implicit RM (+ online would be nice but harder/costlier to setup)

**Questions:**

Is foregoing the partition function Z principled in this setting? This is an important design choice that is not discussed in the current state.

IRPO (Pang et al, 2024) [1] should be mentioned (maybe when discussing iterative DPO)

Authors could draw a parallel between the replay buffer approach and the learning from demonstrations literature (e.g. DQfD, Hester et al, 2017 [2])

What is the granularity of the search for $\alpha^*$ ? What is the cost of conducting such experiments?

Self-rewarding DPO (Yuan et al, 2024) [3] seems very much connected: is it a baseline here? It should be.

[1] - Iterative Reasoning Preference Optimization, arxiv preprint, Pang et al, 2024

[2] - Deep Q-learning from Demonstrations, AAAI, Hester et al, 2017

[3] - Self-Rewarding Language Models, arxiv preprint, Yuan et al, 2024

---

> ### Author Response · Authors · 2024-11-20
> **Rebuttal by Authors [1/3]**
>
> Thank you for your valuable review and suggestions. Below we respond to the comments in weaknesses (***W***) and questions (***Q***).
>
> > Note: **We've marked "3Zep" next to relevant changes in the revised paper to make it easier for you to navigate.**
>
>
> ---
> ***W1. Clarify the connection to prior work.***
>
> Thank you for raising this concern. We appreciate the opportunity to clarify the connection to Park et al., 2024, and acknowledge their significant contributions. Following your suggestion, we have revised our paper to ensure proper attribution, with all updates clearly highlighted in blue.
>
> Both our work and Park et al. share a common idea of mitigating length bias by penalizing verbose responses through regularization—a concept previously explored in the classical RLHF pipeline [Singhal et al.]. Building on this, Park et al. integrate length regularization into the training objectives of direct alignment algorithms such as DPO, while our work applies it at the **data construction stage** in the iterative DPO framework. This distinction offers us an advantage in efficiency. Determining the best regularization coefficient in training objectives often requires multiple training runs, whereas our method can adaptively determine an optimal value for $\\alpha$ in a minute, as acknowledged in the Strengths section of reviewer 3RHj.
>
> [Singhal et al.] A Long Way to Go: Investigating Length Correlations in RLHF
>
> ---
> ***W2. Include regularized DPO as a baseline.***
>
> Following your suggestion, we conducted a comparison between the regularized DPO and our proposed approach, with results presented in $\\textrm{\\color{blue}Table. 7}$. The findings indicate that our approach achieves better performance while effectively mitigating the length exploitation. In contrast, regularized DPO exhibits a quality-length trade-off, where larger regularization coefficients successfully control response length but lead to a decline in response quality.
>
> ---
> ***W3. Further discussion on the results that improvement does not keep over 2 iterations.***
>
> To investigate model performance over three iterations, we compared our approach with the LLM-as-a-Judge baseline (the strongest baseline) on the AlpacaEval 2 benchmark. The results are presented in $\\textrm{\\color{blue}Table. 6}$ of the paper revision.
>
> The table reveals a decline in LC win rates at the third iteration for both our approach and the LLM-as-a-Judge baseline. This aligns with a known challenge in the literature: both self-rewarding language models and SPPO demonstrate consistent improvement only up to three iterations, which corresponds to two iterations in our approach, as our method begins after one initial iteration of DPO training.
>
> We agree with the reviewer's comment that the performance drop in the third iteration may result from the policy that is becoming deterministic over iterations - reduced diversity in preference tuning is a known issue [Kirk et al.]. This reduction in response diversity has negative effects on DPO training [Pal et al.]. It would be an interesting and valuable direction for future work that studies if sampling new responses with a very high temperature can lead to improvement over 3 iterations.
>
>
> [Kirk et al.] Understanding the Effects of RLHF on LLM Generalisation and Diversity
>
> [Pal et al.] Smaug: Fixing Failure Modes of Preference Optimisation with DPO-Positive
>
> ---
>
> ***W4. Mathematical notations and statements.***
>
> > I do not understand why using $\\hat{r}$ instead of $r\^*$ (this is the optimal reward!) l. 34
>
> We agree that $r\^*$ is a more standard notation for the (optimal) reward oracle, and we have updated it accordingly.
>
> > “For a prompt x, we denote its optimal response as $y\^⋆$” --> in general there can be multiple optimal responses (as there can be multiple optimal policies), please adapt the statement & but here authors mean preferred response? this really needs clarification
>
> We would like to clarify that $y\^\\star$ denotes the optimal response, not the preferred response. The optimality is with respect to the optimal reward $r\^\\star$, such that $y\^\\star \\in \\arg\\max\_{y} r\^\\star(y,x)$. For simplicity, we assume that there is a single optimal response with respect to the optimal reward, i.e., $y\^\\star = \\arg\\max\_{y} r\^\\star(y,x)$. Note that the optimal policy is uniquely determined by the optimal reward $r\^\\star$ (up to an additive constant), and their relationship is specified by the equation in Line 34 of the Introduction section. We hope these clarifications could address your question and we have updated accordingly in our revised paper.

---

> ### Author Response · Authors · 2024-11-20
> **Rebuttal by Authors [2/3]**
>
> ***W5. Writing & Clarity.***
>
> Thank you for your careful reading and feedback! We have thoroughly revised our paper to address typos and improve overall clarity.
>
> > Section 4.1 as a whole is really confusing in the current state.
>
> We reorganized the content by moving Section 4.1 in the main text (from the original submission) to Appendix A, for better logical flow and readability.
>
> > citation missing for RLHF (Christiano et al) l. 28
>
> We have added it.
>
> > when mentioning the length exploitation issue, authors could quickly explain why the models exploit length (which is because longer responses are correlated with higher probability of being preferred by human or machine-based preferences)
>
> We have revised the paper (Line 220).
>
>
> > no citation for Gemini Pro! also the version used should be stated (1.0 vs 1.5 vs 1.5 002)
>
> It should be Gemini Pro 1.0, whose result is available on the AlpacaEval 2 leaderboard. We have revised the paper to make it clear.
>
>
> > “Offline DPO w/ new ref: similar to offline DPO but we assign the current policy as the new reference model, while we use a fixed reference model in Offline DPO. This corresponds to γ = 1” -> misleading as my understanding is that the reference is fixed (it is just the end policy of the first round of DPO)
>
> Yes, the reference policy should be fixed during a single round of DPO training. Here, after each round of iteration, we update the reference policy with the latest trained model, and proceed with another round of training. We have revised the paper to improve the clarity.
>
>
> ---
>
> ***W6. Experiments/Ablation that can be added.***
>
> > What about using all the K samples with a Plackett-Luce model? Does this bring any benefit?
>
> Incorporating more samples in preference tuning with a Plackett-Luce model could potentially enhance performance, as these samples provide more fine-grained information. Prior work by Song et al. has explored this direction and demonstrated promising results. Due to the time limit during rebuttal, we leave a thorough investigation of combining K samples with DICE as future work.
>
> > Comparing replaying human preference data with KL regularization to the previous policy.
>
> We acknowledge that KL regularization targets the same purpose as experience replay - avoiding forgetting the knowledge inbuilt in the initial policy. This can be done by incorporating one additional KL term in the RLHF objective:
> $$
> \\max \_{\\pi\_\\theta} \\mathbb{E}\_{\\mathbf{x} \\sim \\mathcal{D}, \\mathbf{y} \\sim \\pi\_\\theta(\\mathbf{y} | \\mathbf{x})} {\\left[r(\\mathbf{x}, \\mathbf{y})\\right]}
> -\\beta \\cdot\\mathbb{D}\_{\\mathrm{KL}}\\left[\\pi\_\\theta(\\mathbf{y} | \\mathbf{x}) \\| \\pi\_{\\textrm{ref}}(\\mathbf{y} | \\mathbf{x})\\right]-\\beta \\cdot\\mathbb{D}\_{\\mathrm{KL}}\\left[\\pi\_\\theta(\\mathbf{y} | \\mathbf{x}) \\| \\pi\_{\\theta\^{(0)}}(\\mathbf{y} | \\mathbf{x})\\right]\\textrm{,}
> $$
>
> Following the derivation of DPO, a similar contrastive loss can be derived. Yet, this approach faces practical issues: one extra policy model $\\pi\_\\theta\^{(0)}$ needs to be loaded on GPUs, substantially increasing the memory overhead during training. Compared with this technique, experience replay is a more lightweight and stable alternative.
>
> > Baseline missing: offline AI feedback annotations instead of implicit RM.
>
> In our experiment, we included a baseline that utilized AI feedback via LLM-as-a-judge prompting, referred to as the "LLM-as-a-Judge" baseline method in $\\textrm{\\color{blue}Table. 1}$.
>
>
> [song et al.] Preference ranking optimization for human alignment. AAAI-24.

---

> ### Author Response · Authors · 2024-11-20
> **Rebuttal by Authors [3/3]**
>
> ***Q1. Is foregoing the partition function Z principled in this setting?***
>
> It depends on how we use the DPO implicit reward. In our case, the absolute values of the implicit rewards are not important, because we only need the ranking of two responses. This is done by taking the difference between the implicit rewards of two responses, where the partition function simply cancels because it depends on $x$ only. Therefore, we argue foregoing the partition function $Z$ in our case is proper.
>
> ---
>
> ***Q2 & Q3. IRPO (Pang et al, 2024) should be mentioned. Authors could draw a parallel between the replay buffer approach and the learning from demonstrations literature (e.g. DQfD, Hester et al, 2017)***
>
> We have addressed both in the revised paper (Line 292).
>
> ---
>
>
> ***Q4. What is the granularity of the search for $\\alpha\^\\star$? What is the cost of conducting such experiments?***
>
> We adopt a simple grid search with step size being 0.001. It runs for ~33 seconds on a CPU-only laptop.
>
> ---
>
> ***Q5. Self-rewarding DPO (Yuan et al, 2024) [3] seems very much connected: is it a baseline here? It should be.***
>
> Yes, we have included self-rewarding DPO in our comparison. Specifically, we implement their core idea, which is to use LLM-as-a-judge to score individual responses so as to construct a preference dataset without the need for external reward models, and conduct apple-to-apple comparison with our method. Self-rewarding DPO corresponds to the "LLM-as-a-Judge" baseline method in $\\textrm{\\color{blue}Table. 1}$.

---

> > ### Comment · Reviewer_3Zep · 2024-11-25
> > **Update on Rebuttal [3Zep]**
> >
> > I raised my score significantly after authors addressed major concerns of mine. I thank authors for the diligent work on making the submission better and fostering proper credit assignment.

---

> > > ### Author Response · Authors · 2024-11-25
> > > **Thank you for your support**
> > >
> > > Thank you for your support and for raising the score! We greatly appreciate your detailed feedback and valuable suggestions, which have been instrumental in improving our work. In the final revision, we will further polish our paper to incorporate the insights from the rebuttal discussions. Thank you again!

---

### Official Review · Reviewer_A3Uc · 2024-11-01

**Soundness:** 2
**Presentation:** 3
**Contribution:** 2
**Rating:** 5
**Confidence:** 3

**Summary:**

This paper proposes a self-supervised iterative finetuning procedure for LMs trained with DPO, which, after the first round of DPO (with human data), uses the Implicit Reward Model (as used by DPO) to label new self-supervised preference data, and then trains again. This process can be repeated. To obtain good performance, the authors propose 2 techniques that improve the nth stage of finetuning and show that these techniques are critical for thei good results: adjusting the Implicit Rewards by a length penalty, and using an experience replay to mix in the human labeled data. The authors find that this iterative procedure obtains better results than several baselines on AlpacaEval and ArenaHard.

**Strengths:**

Iterative finetuning on self-supervised data is an important direction, which has received a reasonable amount of attention, e.g., with Anthropic's Constitutional AI / RLAIF, and others. The DPO approach has also gained significant adoption, so improving DPO with iterative finetuning would be of interest to the community.

The proposed procedure is straightforward and the results are good. The paper as a whole is well written and presented. I believe I could reimplement the method/experiments based on the presented details. The implementation is novel, as is the use of implicit rewards for iterative finetuning as far as I am aware.

**Weaknesses:**

Main concern:
- I remain unconvinced this is the best approach to iterative finetuning a DPO model, since the evaluation only tests downstream performance and other potential reward models that are known to be empirically stronger than implicit DPO rewards are ignored: If you look at RewardBench, the implicit rewards of DPO models are quite far down the list in terms of their strength as a reward model. Furthermore, reward models themselves are not very difficult to train, so the overhead as compared to using the implicit DPO reward is not that large. See also Lin et al. 2024, On the Limited Generalization Capability of the Implicit Reward Model...
So for these reasons, my prior that using a trained RM for iteratively improving a DPO model (e.g., as done by He et al. 2024, Semi-Supervised Reward Modeling via Iterative Self-Training...) is unchanged after reading this paper... i.e., I do not think the proposed approach is competitive with simple alternatives, which do not appear as baselines.
To convince me here (or alternatively, to contribute a "full" empirical study that determines what is still an open question in my mind), I would want to see the iterative DPO w/ trained RM show up in Table 1, OR alternatively, I would want to see that the implicit reward model that is learned via this iterative process (and adjusted with the LC normalization) is competitive vs other RMs on RewardBench or similar.
Side note: esp. given the length regularizer, avg log prob reward (see simpo) would be a good baseline to have

Concern:
- Limited Conceptual Novelty: While the use of implicit rewards for iterative finetuning is new, neither iterative finetuning, nor the implicit reward model are new concepts... and it is in some sense "obvious" that one can use any reward model for iterative finetuning, including the implicit reward model. Therefore, the main novelty in my view is limited to (a) empirical results (which are good, but see point above), and (b) the length-normalization penalty (although others have also investigated different variations of this) and experience replay (not sure I would call this new, but the empirical demonstration of its usefulness in this context is good).

Lesser concern:
- I think Section 4.1 detracts from the paper, rather than adds to it. E.g., Equation (6) seems wrong to me (unless you mean S to be ALL possible completions besides the "optimal" one, but this is also wrong, because "optimal" is ill defined here (the optimal response out of all possible completions under the implicit reward will likely be overoptimized / OOD / poor). I'm not sure this adds much intuition, and perhaps even supports the wrong intuitions (e.g., just because a high likelihood suboptimal response is not sampled, that doesn't mean it's probability doesn't change... and this criticism applies to finite on-policy data as well).

Minor/Irrelevant to score:
- I would call the length penalty reward specification/augmentation/engineering rather than reward shaping, which in my experience is reserved for rewards that are meant to improve the learning process but maintain the same desired policy (see Ng et al. 1999, Policy Invariance...)
- Tortured acronym that is already used elsewhere. Consider something more unique that actually maps to the method name, e.g., Iterative Finetuning with Implicit Reward, or if you insist on a "word", Finetuning with Implicit REwards gives you FIRE, so you can do FFIRE (Finetuning Further) or IFIRE (Iterative FIRE).

**Questions:**

- If there is anything the authors can do to convince me that using the adjusted DPO implicit rewards has an advantage over training a separate reward model for iterative finetuning, this would be greatly appreciated (see Weakness #1 above for some ideas). (I am aware that using the DPO implicit reward is computationally more efficient, but I do not believe that advantage alone is sufficient, at least not without fully understanding the compute-return trade-off here).

---

> ### Author Response · Authors · 2024-11-20
> **Rebuttal by Authors [1/2]**
>
> Thank you for your valuable review and suggestions. We appreciate the recognition of iterative training with self-supervised data, also named self-alignment, as an important research direction. Below we respond to the comments in weaknesses (***W***) and questions (***Q***).
>
>
> > Note: **We've marked "A3Uc" next to relevant changes in the revised paper to make it easier for you to navigate.**
>
>
> ---
> ***W1 & Q1. Experiments lack of comparison with other external reward model.***
>
> Thank you for raising this question. We would like to clarify that our work adheres to the self-alignment setup, which focuses on improving LLMs without external supervision [Yuan et al., Sun et al, Gao et al., Zhang et al.] (e.g., powerful reward models from RewardBench). This setup is particularly important because, in some scenarios, obtaining an external reward model is challenging such as for models tailored to specialized domains (e.g., medical or legal language), where a general-purpose reward model may not be suitable [Colombo et al.]. Therefore, investigating methods for improving language models without external reward models remains valuable, as also highlighted in [this official comment](https://openreview.net/forum?id=dliIIodM6b&noteId=tm86jY6HAQ) of Reviewer **3RHj**.
>
> Since our method relies on an initial round of DPO with human preference data (i.e., the seed data), we acknowledge that a relevant baseline would involve first training an **internal reward model (IntlRM)** using the same seed data and then leveraging this IntlRM for iterative improvement. This comparison allows us to evaluate the effectiveness of our DPO implicit reward against a dedicated reward model within the self-alignment setup.
>
> The IntlRM training follows the standard setting of OpenRLHF [Hu et al.], and we have included the training scripts in the updated Supplementary Material for reproducibility. The following table compares the performance of our DPO implicit reward and IntlRM:
>
> | Method | Alignment Rate |
> |-----|---|
> | IntlRM | 0.624 |
> | ERM-555k  | 0.656
> | DPO Implicit Reward | **0.698** |
>
>
> The experiment results indicate that the DPO implicit reward achieves higher accuracy than the IntlRM trained on the same preference data. Furthermore, it surpasses the performance of an external reward model, ERM-555k, trained on substantially more data. This implies that the implicit reward model offers distinct advantages when evaluating its own generated data, though the scalar reward models in general excel on a wider range of tasks when the preference data is abundant, as demonstrated by ArmoRM-Llama3-8B-v0.1, which was trained on 1M preference data. Based on these findings, we argue that the implicit reward is a competitive option in the self-alignment setting. For further details, refer to $\\textrm{\\color{blue}Section 5.4}$ and $\\textrm{\\color{blue}Table. 5}$ in the paper revision.
>
>
> [Yuan et al.] Self-Rewarding Language Models
>
> [Sun et al.] SALMON: Self-Alignment with Instructable Reward Models
>
> [Gao et al.] Human-Instruction-Free LLM Self-Alignment with Limited Samples
>
> [Zhang et al.] Self-Alignment for Factuality: Mitigating Hallucinations in LLMs via Self-Evaluation
>
> [Colombo et al.] Saullm-7b: A pioneering large language model for law.
>
> [Hu et al.] OpenRLHF: An Easy-to-use, Scalable and High-performance RLHF Framework

---

> ### Author Response · Authors · 2024-11-20
> **Rebuttal by Authors [2/2]**
>
> ***W2. Limited conceptual novelty.***
>
> Although the implicit reward has been acknowledged in prior works, our study investigates the underexplored question of **whether the implicit reward model can be effectively utilized to further improve the LLM itself**, and we answer the question in the affirmative. We believe our findings offer a fresh perspective to the self-alignment literature and, more importantly, it provides special advantages in the scenario where the general-purpose scalar reward model fails to generalize (see discussion in ***W1***).
>
> Regarding the length regularization, we would like to clarify that while prior works [Meng et al., Park et al.] in LLM alignment have investigated variants of length regularization, these approaches typically incorporate the regularization term into the training loss function. However, such approaches often require extensive hypertuning (e.g., involving multiple training runs that may take hours). In contrast, our approach applies reward shaping before DPO training to construct the preference data, which can be completed within a minute.
>
> [Meng et al.] Simpo: Simple preference optimization with a reference-free reward.
>
> [Park et al.] Disentangling length from quality in direct preference optimization.
>
> ---
>
> ***W3. Clarification on section 4.1.***
>
> Following your suggestion, we have moved the original Section 4.1 to $\\textrm{\\color{blue}Appendix. A}$ to improve the flow of the revised paper. Below, we provide clarifications on your questions.
>
> > Equation (6) seems wrong to me unless you mean S to be ALL possible completions besides the "optimal" one
>
> Yes, $\\mathcal{S}$ refers to the set of all suboptimal responses. The main purpose of this analysis is to give intuitions on why **on-policy sampling** could benefit iterative DPO training, which we employ to build our self-alignment pipeline. Here, we assume the "optimality" of responses is measured according to the oracle reward, instead of the learned DPO implicit reward, so there should not be over-optimization or OOD issues. Nevertheless, we admit this section may cause misunderstanding, especially when put in our method section. We have moved it to $\\textrm{\\color{blue}Appendix. A}$ and added necessary footnotes.
>
> ---
>
> ***W4. Minor concerns***
>
> > I would call the length penalty reward specification/augmentation/engineering rather than reward shaping, which in my experience is reserved for rewards that are meant to improve the learning process but maintain the same desired policy.
>
> Thank you for your insightful comment. While we used the term "reward shaping" to describe our method of adaptively adjusting the reward based on response length, we acknowledge its more specific usage in the literature. To avoid confusion during the rebuttal, we will retain the term for now but will consider adopting a more precise term, such as "reward specification" suggested by the reviewer, in a future revision.
>
> > Tortured acronym that is already used elsewhere. Consider something more unique that actually maps to the method name.
>
> Thank you for highlighting the overlap and suggesting alternatives. We will explore a more unique and descriptive acronym for a future revision.

---

> ### Author Response · Authors · 2024-11-25
> **Looking forward to further feedback**
>
> Dear Reviewer A3Uc,
>
> Thank you once again for your constructive feedback. We would like to kindly remind you that we have included comparisons with other reward models in $\\textrm{\\color{blue}Section 5. 4}$ and $\\textrm{\\color{blue}Table. 5}$ of the paper revision. Additionally, we have clarified our contributions and improved the overall writing of the paper.
>
> ---
>
> As the discussion period is coming to a close in two days, we look forward to your response and would be happy to address any further comments or questions you may have.
>
> Best,
>
> The Authors

---

> > ### Comment · Reviewer_A3Uc · 2024-11-26
> >
> > I like the new Section 5.4, and the results are encouraging. I also like the argument about special use cases where general purpose RMs are not available. And I appreciate moving the 4.1 to the Appendix.
> >
> > To make sure I am understanding correctly:
> > - You are training the RM from the base sft model on the offline Ultrafeedback.
> > - Does the "DPO Implicit Reward" use the implicit rewards from the base DPO Zephyr, or from a version that has been through DICE on self generated data with GPT 4o labels? (it would be good to make this extra clear in the text)
> >
> > If the latter, this comparison seems a bit unfair (I would be interested to see the RM tuned on the same data as the DPO Implicit Reward). I will likely raise my score anyways.

---

> > > ### Author Response · Authors · 2024-11-27
> > > **Thank you for the encouraging feedback!**
> > >
> > > Thank you for the encouraging feedback! Below, we clarify the two questions regarding the experimental setup:
> > > - Yes, the RM (IntlRM) is trained from a base SFT model on the offline Ultrafeedback. Specifically, both the "DPO Implicit Reward" (with the policy model *zephyr-7B-beta* and the reference model *mistral-7b-sft-beta*) and IntlRM are **trained from the same base model (mistral-7b-sft-beta) on the same dataset (Ultrafeedback)**.
> > > - The "DPO Implicit Reward" uses the implicit rewards derived from the base DPO Zephyr. Labels from GPT-4o are generated exclusively for the 500 test cases and are not used during training.
> > >
> > > We have updated $\\textrm{\\color{blue}Section 5. 4}$ in the lastest paper revision to clarify the experimental setup.

---

> > > ### Author Response · Authors · 2024-11-30
> > > **Looking forward to further updates**
> > >
> > > Dear Reviewer A3Uc, as the discussion period is nearing its end, we would like to kindly follow up to see if there are any additional clarifications or information needed from our side. We look forward to any updates you may have. Thank you again for your valuable insights!

---

### Official Review · Reviewer_3RHj · 2024-11-01

**Soundness:** 2
**Presentation:** 3
**Contribution:** 4
**Rating:** 8
**Confidence:** 4

**Summary:**

The paper introduces a new approach to iterative DPO. After each iteration, synthetic comparisons are generated by the current policy: given a prompt, 16 completions are sampled and then labeled using the implicit reward from the DPO formalism; a length penalty is applied to make the comparison dataset approximately length-unbiased; and the best and worst completions are used for the synthetic comparison. These synthetic comparisons are then used for the subsequent iteration. The method outperforms baselines on two different benchmarks across two different models.

**Strengths:**

The method is natural and straightforward to apply: using the current policy to generate both new on-policy samples and rewards provides and easy way to do iterative policy improvement with no additional models or data. I expect the method to be of great interest to practitioners who value this benefit, which is similar to the benefit of DPO compared to PPO for RLHF.

The experiments are thorough and promising enough that I expect others will want to try out the method in their own settings, although I discuss a major weakness below. The ablations are also well-chosen and interesting. In particular, the ablation for the choice of alpha* mostly satisfied me that there is a real benefit to the method beyond the length penalty, although see my question 2 on this point.

I thought the hypothesis about on-policy data helping was a good one, but I would suggest an additional hypothesis for why the method helps, namely that using the implicit rewards is similar to training a reward model, so as with PPO there may be a benefit from discrimination being easier than generation.

**Weaknesses:**

A glaring methodological weakness is in the treatment of the hyperparameter gamma (1 minus the fraction of offline data used): if I am understanding correctly, the tuning was done using the final reported results. This is contrary to standard machine learning best practices, whereby hyperparameters are tuned using a validation set. This gives the method a significant advantage if the difference between different values of gamma has a significant noise component. The same applies to the hyperparameter beta, but the issue is not as important there because the same tuning was done for the baselines too (although I still think this should be fixed – in particular, the Base model does not get to tune beta). Regardless, some clarification should be given as to how the tuning for beta was done (is it the same across models/methods/iterations/benchmarks?). I do not believe the hyperparameter K was tuned, but it could be good to clarify how K = 16 was chosen.

It looks like the same value of gamma was used across benchmarks, which is good since this reduces the selection effect, but different values of gamma were used between different models (and moreover, this fact is not clearly disclosed, I attempted to infer this by comparing Table 1 with Appendix C). Since it would take a lot of work to re-run all the experiments with fairly-tuned hyperparameters, my suggested remedy is as follows: **report results for the same value of gamma (say gamma = 0.5) across all experiments from both models**. I think this would go a long way towards presenting a fairer picture of the algorithm's performance compared to other methods.

**I have rated the paper and 8 in spite of this scientific mis-step in anticipation of some sort of correction or explanation.** Without the above suggested change or some other correction or explanation, I may lower my score, which would be a shame, because it is otherwise a great paper. Similarly, I may also raise my score to a 10 if I become convinced that this problem has been adequately addressed.

Another limitation is a lack of comparison to reward model-based methods such as PPO, but I think it is reasonable to leave this to future work. I think there are some interesting comparisons to be made here because the use of the implicit rewards is similar to training a reward model and using the rewards from that.

The presentation was generally pretty clear, although I found Algorithm 1 somewhat hard to follow (and it would be even harder for someone glancing at the paper), for a couple of reasons:
- It seems worth stating that the comparison dataset is created by taking the best and worst samples according to the (length-penalized) reward.
- I believe there is a typo where it says to use "pi_ref^(t-1) as reference policy", and it should be pi_ref^(t) (unless I am missing some subtlety that deserves further explanation). But since pi_ref^(t) is equal to pi_{theta^(t-1)}, I think it would be simpler and clearer to remove the notation pi_ref^(t) from the pseudocode entirely.

Minor line-by-line comments:
- Abstract: "refinements that debias the length of the responses and enhance the quality of the preference dataset" - this phrase is convoluted and unclear, suggest replacing with "a length penalty to make the preference dataset length-unbiased" or just "a length penalty" may suffice.
- Line 046: typo "by itself" -> "by DPO itself"
- Line 216: might be good to clarify where does pi_mu comes from in practice
- Line 227: "without minimizing the likelihood of other suboptimal responses" - I didn't understand what point you were trying to make, since the loss doesn't depend on these other responses by definition?
- Line 245: "subtrahend" - I had never seen this word before but it made sense after I Googled it :)
- Line 291.5: "normally distributed" - It does not look normally distributed at all, there is a big spike at 0. It looks more like a Laplace distribution (although I wouldn't want to make this claim without statistical evidence, and in any case it doesn't seem important to the exposition).
- Equation (9) - It would seem more natural to put the absolute value outside the expectation instead of inside, since this would directly optimize for the dataset being unbiased in the sense of the mean length of winning completions roughly matching the mean length of losing completions. That said, this probably gives a very similar value of alpha* and so this change would mostly be for improved theoretical soundness.
- Tables 3 and 4: I assume this is for AlpacaEval 2, but maybe good to clarify in the table captions.

**Questions:**

1. Are the base models both trained on the same UltraFeedback dataset from which the offline dataset is sampled? It seems likely yes, but it would be good to clarify this in Section 5.1, since it affects how to interpret the comparison with the base and offline DPO models.
2. What is the length bias like for LLM-as-a-Judge? If it is too large, an ablation that uses a length penalty with this method would seem appropriate.

---

> ### Author Response · Authors · 2024-11-20
> **Rebuttal by Authors [1/2]**
>
> Thank you for your encouraging review, valuable suggestions, and additional supportive comments. Below we address the points raised under **Weaknesses (W)** and **Questions (Q)**.
>
> > Note: **We've marked "3RHj" next to relevant changes in the revised paper to make it easier for you to navigate.**
>
> ---
>
> ***W1. Clarification on hyperparameter tuning.***
>
> Thank you for pointing this out. In our experiments, the hyperparameters, including $\\gamma$ and $\\beta$, were selected based on the model performance on AlpacaEval 2 (AE2). These parameters were tuned for each method and model separately (with $\\gamma$ specific to our approach) but remained fixed across iterations. Specifically, $\\beta$ was tuned during the first iteration from the set {0.01, 0.1}, while $\\gamma$ was tuned using the ranges {0.0, 0.25, 0.50, 0.75, 1.0} for the Zephyr backbone and {0.0, 0.10, 0.25, 0.50, 0.75, 1.0} for the Llama-3 backbone. The parameter $K$ was fixed at 16 for simplicity in implementation - we could generate all the responses in 2 shots on an 8-GPU server using data parallelism. The best hyperparameter setting identified in AlpacaEval 2 was directly applied in Arena-Hard (AH) for all methods and models without any further tuning.
>
> We agree with the reviewer that using a validation set for hyperparameter tuning would be more rigorous. Following the best practice, hyperparameters should be optimized on a validation set, and the model's performance should be subsequently evaluated on a separate test set or benchmark. As it would take a long time to construct a validation set/benchmark and conduct the whole pipeline with thoroughly tuned hyperparameters, we try to address this concern by treating the two benchmarks (AE2 and AH) as proxies for validation and test sets. Specifically, we validate on AE2 while testing on AH, and then reverse the roles of the two benchmarks.
>
> Below we show the results in $\\textrm{\\color{blue}Table. R2.1}$ and $\\textrm{\\color{blue}Table. R2.2}$:
>
> $\\textrm{\\color{blue}Table. R2.1}$ Model performance on AlpacaEval 2 (AE2) and Arena-Hard (AH) across different $\\gamma$ for the Zephyr Backbone. LC denotes the length-controlled win rate. CI denotes confidence interval.
>
> | $\\gamma$ | LC. AE2 | LC.AH: Score (95% CI) |
> |---|---|---|
> | 0.00 | 18.88 | 17.6 (-1.4, 1.7) |
> | 0.25 | 17.87 | 16.1 (-1.2, 1.4) |
> | **0.50** | **20.71** | **18.4** (-1.3, 1.6) |
> | 0.75 | 16.90 | 14.9 (-1.6, 1.4) |
> | 1.00 | 13.4 | 13.0 (-1.3, 1.3) |
>
>
> $\\textrm{\\color{blue}Table. R2.2}$ Model performance on AlpacaEval 2 (AE2) and Arena-Hard (AH) across different $\\gamma$ for the Llama-3 Backbone. LC denotes the length-controlled win rate. CI denotes confidence interval.
>
> | $\\gamma$ | LC. AE2 | LC.AH: Score (95% CI) |
> |---|---|---|
> | 0.00 | 26.93 | 39.1 (-2.3, 2.5) |
> | **0.10** | **27.55** | **39.1** (-2.0, 2.4) |
> | 0.25 | 24.78 | 33.3 (-2.1, 1.8) |
> | 0.50 | 23.59 | 32.2 (-2.3, 2.2) |
> | 0.75 | 22.57 | 27.4 (-2.0, 1.8) |
> | 1.00 | 22.5 | 24.5 (-1.7, 1.8) |
>
> The two benchmarks and two backbone models constitute four validation-test instances. We observe that the $\\gamma$ selected from the validation set/benchmark consistently achieves the best performance on the test set across all instances, demonstrating the optimal value of $\\gamma$ is robust to the choice of the validation set/benchmark. Meanwhile, we do observe that the optimal $\\gamma$ for different backbone models is varied, which is expected, as different backbones produce responses of varying quality. Higher-quality generated data reduces the need for replayed experience, aligning with our observations for the Llama-3 backbone.
>
> ---
> ***W2. Comparison to reward model-based approach like PPO.***
>
> Thank you for suggesting this relevant comparison. It would be interesting to explore how the implicit reward model interacts with PPO. We are actively working on these comparisons and will provide a follow-up comment once the experiments are completed.

---

> ### Author Response · Authors · 2024-11-20
> **Rebuttal by Authors [2/2]**
>
> ***W3. Line-by-line comments***
> > Algorithm 1: worth stating that the dataset is created by taking the best and worst samples according to the (length-penalized) reward
>
> We have updated accordingly.
>
> > Algorithm 1: I believe there is a typo where it says to use "pi\_ref\^(t-1) as reference policy", and it should be pi\_ref\^(t)
>
> This is the last comment after line 4 in Algorithm 1, which specifies the target ($\\pi\_\\theta$) and reference ($\\pi\_{\\mathrm{ref}}$) policies which we use to evaluate the length-regularized reward:
> $r\_{\\mathrm{LR}}(x,y;\\alpha) = \\beta \\log \\frac{\\pi\_\\theta(y|x)}{\\pi\_{\\mathrm{ref}}(y|x)} - \\alpha |y|$. Suppose we are constructing the dataset for round $t$, we need to ensure that $\\pi\_\\theta$ is the trained model after round $t-1$, and that $\\pi\_{\\mathrm{ref}}$ is the reference model used in round $t-1$. This is to ensure that the DPO training in round $t-1$ makes the first term in $r\_{\\mathrm{LR}}$ a valid implicit reward to be used in round $t$.
>
> Therefore, we would like to clarify that this is not a typo. In Algorithm 1, taking $t=1$ as an example, $r\_{\\mathrm{LR}}$ is obtained with $\\pi\_{\\theta\_0}$ and $\\pi\_{\\mathrm{ref}}\^{0}$, which are exactly the initial DPO-tuned policy and the initial reference policy (used to tune $\\pi\_{\\theta\_0}$) given in the algorithm input (as well as in $\\textrm{\\color{blue}line 201}$).
>
> > Suggestions to modify the abstract
>
> Thanks for this suggestion and we have updated it accordingly.
>
> > Line 216: might be good to clarify where does pi\_mu comes from in practice
>
> $\\pi\_\\mu$ is the behavior policy that collects the offline dataset, which can be a mixture of existing LLMs or human experts. We updated with a footnote in the revised paper (the original Section 4.1 is moved to Appendix A for better logical flow and readability).
>
> > Line 227: "without minimizing the likelihood of other suboptimal responses" - I didn't understand what point you were trying to make, since the loss doesn't depend on these other responses by definition?
>
> Yes, given the offline dataset, the DPO loss can be minimized to zero by only minimizing the likelihood of the losing response appearing in the dataset, while other suboptimal responses out of the dataset can still remain high likelihood.
>
> > Equation (9) - It would seem more natural to put the absolute value outside the expectation instead of inside
>
> Thank you for this insightful point. We apologize that the original Eq.9 (now Eq. 6 in the revised paper) contains a typo. We have carefully checked our implementation and confirmed that we actually took the absolute value outside the expectation. So all our experiments use the version that you have kindly described. Running the alternative gives 0.021, which is close to 0.023. We have fixed the typo in Eq.6.
>
>
> ---
> ***Q1. Are the base models both trained on the same UltraFeedback dataset from which the offline dataset is sampled?***
>
> Yes. We have clarified this in Section 5.1. Thank you for your suggestion.
>
> ---
> ***Q2. What is the length bias like for LLM-as-a-Judge?***
>
> We refer to $\\textrm{\\color{blue}Figure. 4 in Appendix C}$ for the average absolute length difference with both our implicit reward model and the LLM-as-a-judge reward model. The figures plot the optimization landscapes of Eq. 6, where we observe that the optimal $\\alpha$ is $0$ for LLM-as-a-judge, indicating that it seems not to cause length bias. Therefore, we do not apply a similar technique to post-processing the preference dataset for LLM-as-a-Judge baseline.

---

> > ### Comment · Reviewer_3RHj · 2024-11-25
> > **Response to rebuttal**
> >
> > Thank you for your rebuttal. The cross-validation method for hyperparameter tuning that you suggested seems very appropriate. I didn't see any change to Table 1 in the rebuttal version, but I would be satisfied if the paper is updated for the camera-ready version to use this cross-validation method, and am keeping my score assuming this update is made.
> >
> > Thank you for clarifying that what I thought was a typo in Algorithm 1 is correct. I think this subtlety could be missed by people and suggest changing the prose to mention it, and removing the use of pi_ref^(t) from Algorithm 1 and instead replacing it by pi_{theta^(...)} to reduce notational clutter and improve clarity.

---

> > > ### Author Response · Authors · 2024-11-26
> > > **Thank you for the encouraging feedback!**
> > >
> > > Thank you for the encouraging feedback and the constructive suggestions! We have refined the description of the hypertuning in the **Training Details** paragraph in $\\textrm{\\color{blue}Section 5.1}$ and included additional details in $\\textrm{\\color{blue}Appendix F}$. Since the value of $\\gamma$ selected by cross-validation matches the one used in Table 1, the evaluation results for our approach remain unchanged.
> > >
> > > In response to the suggestion to compare with PPO using implicit rewards, we have conducted these experiments. The results, presented in $\\textrm{\\color{blue}Table 8}$ of the latest revision, indicate that PPO with implicit reward model improves upon the base model (Zephyr-7B-beta). However, the improvement is less pronounced compared to DPO. Detailed experiment settings can be found in $\\textrm{\\color{blue}Appendix D}$.
> > >
> > > Regarding the suggestion to improve the clarity of $\\pi_\\textrm{ref}$, we agree that replacing $\\pi_\\textrm{ref}$ with $\\pi_\\theta$ (with appropriate indexing) would be clearer and easier to follow. However, implementing this change in the current version would necessitate introducing $\\pi_{\\theta^{(-1)}}$, which is rarely seen in standard mathematical notation. Properly addressing this issue would require us to reorganize the notation system of the paper, which we plan to undertake in a future revision to avoid any confusion during the rebuttal phase.
> > >
> > > Once again, we deeply appreciate the reviewer's insightful comments, time, and effort devoted to improving our paper!

---

> ### Comment · Reviewer_3RHj · 2024-11-26
>
> Great, I think these changes improve the paper. I am happy to see that other reviewers have increased their scores, and will continue to advocate for acceptance of the paper. I will keep my score as is since it was already in anticipation of such improvements.

---

> > ### Author Response · Authors · 2024-11-27
> > **Thank you for your support**
> >
> > We appreciate your continued support and valuable insights regarding our paper. We will incorporate the feedback from the rebuttal discussion and further refine our paper to enhance the clarity and quality. Thank you again!

---

### Official Review · Reviewer_6mfr · 2024-11-04

**Soundness:** 3
**Presentation:** 3
**Contribution:** 2
**Rating:** 3
**Confidence:** 4

**Summary:**

This paper introduces a self-alignment approach, DICE, which uses the implicit reward model from DPO in a bootstrapping manner to further align LLMs, achieving over 8% improvement in alignment on AlpacaEval 2 without external feedback.

**Strengths:**

The writing is clear and easy to follow, and the experiments provide support for the claim that repeated use of the implicit reward model could enhance performance.

**Weaknesses:**

The results offer a useful insight, though the approach itself may lack significant novelty, and the improvements seem to be marginal. Additionally, length-regularized reward shaping is already a widely adopted technique. While the paper seeks to improve model performance through Direct Preference Optimization (DPO) by leveraging the implicit reward model, it would benefit from a theoretical explanation of why repeated use of the implicit reward model could enhance performance. Moreover, it might be worth considering whether repeatedly using the implicit reward model offers more advantages than utilizing an additional reward model (RM) to label subsequent iterations.

**Questions:**

N/A

---

> ### Author Response · Authors · 2024-11-20
> **Rebuttal by Authors [1/2]**
>
> Thank you for your valuable review and suggestions. We appreciate that you find our results provide useful insights. Below, we address the comments on **Weaknesses (W)** with empirical evidence.
>
>
> > Note: **We've marked "6mfr" next to relevant changes in the revised paper to make it easier for you to navigate.**
>
>
> ---
>
> ***W1. Proposed approach lacks significant novelty.***
>
> Thank you for raising this concern. We would like to highlight the key contributions that distinguish our work from prior research:
> 1. Our work proposes to use a principled alternative reward model, specifically the implicit reward derived from DPO training, in the self-alignment (self-improving fine-tuning) literature. In contrast, prior approaches primarily rely on prompting the language model for rewards [Yuan et al., Sun et al.] using the LLM-as-a-judge technique;
> 2. In addition, we propose two techniques, length-regularized reward shaping and experience replay, that are shown to be essential for achieving optimal performance with our approach.
>
> Although the implicit reward has been acknowledged in prior works, our study investigates the underexplored question of **whether the implicit reward model can be effectively utilized to further improve the LLM itself**, and we answer the question in the affirmative. We believe our findings offer a fresh perspective to the self-alignment literature and also provide a practical solution for practitioners seeking improvements over the vanilla DPO, as also highlighted in [this official comment](https://openreview.net/forum?id=dliIIodM6b&noteId=tm86jY6HAQ) of Reviewer **3RHj**.
>
> [Yuan et al.] Self-Rewarding Language Models
>
> [Sun et al.] SALMON: Self-Alignment with Instructable Reward Models
>
> ---
>
> ***W2. Improvement of the proposed approach is marginal.***
>
> To address your concerns about the extent of improvement, we present both the absolute and relative improvements in LC (length-controlled win rate) on AlpacaEval 2 (AE2) and WR (win rate) on Arena-Hard (AH). These improvements are summarized in the tables below:
>
> $\\textrm{\\color{blue}Table. R1.1}$ Absolute and relative improvements of LC on AE2 and WR on AH for the Zephyr Backbone.
>
> |  | **Raw LC.AE2** | **Absolute LC.AE2** | **Relative LC.AE2** | **Raw WR.AH** | **Absolute WR.AH** | **Relative WR.AH** |
> |---|---|---|---|---|---|---|
> | Base | 12.69 | - | - | 9.9 | - | - |
> | Highest Baseline | 15.36 | 2.67 | 21.04% | 15.7 | 5.8 | 58.59% |
> | DICE | 20.71 | 8.02 | 63.20% | 16.7 | 6.8 | 68.69% |
>
>
> $\\textrm{\\color{blue}Table. R1.2}$ Absolute and relative improvements of LC on AE2 and WR on AH for the Llama-3 Backbone.
>
> |  | **Raw LC.AE2** | **Absolute LC.AE2** | **Relative LC.AE2** | **Raw WR.AH** | **Absolute WR.AH** | **Relative WR.AH** |
> |---|---|---|---|---|---|---|
> | Base | 18.20 | - | - | 21.6 | - | - |
> | Highest Baseline | 22.50 | 4.30 | 23.63% | 28.7 | 7.10 | 32.87% |
> | DICE | 27.55 | 9.35 | 51.37% | 41.2 | 19.60 | 90.74% |
>
> The values are taken or derived from $\\textrm{\\color{blue}Table. 1}$ in the paper. **Absolute** improvement refers to the direct difference in performance metrics between any evaluated method (e.g., our approach or a baseline) and the base model. **Relative improvement** is calculated as the absolute improvement divided by the base model's score. The `Highest Baseline` refers to the best score across all baselines in $\\textrm{\\color{blue}Table. 1}$. Across all benchmarks and backbones, our approach demonstrates substantial improvements (as also acknowledged by all the other three reviewers), with minimal relative gains of 51.37% on AE2 and 68.69% on AH. Notably, our method consistently outperforms the best baseline by a significant margin.

---

> ### Author Response · Authors · 2024-11-20
> **Rebuttal by Authors [2/2]**
>
> ***W3. Length-regularized reward shaping is a widely adopted technique.***
>
> While prior works in LLM alignment have explored variants of length regularization, these approaches typically integrate the regularization term into the training objectives. In contrast, our approach applies it at the **data construction stage** in the iterative DPO framework. This distinction offers us an advantage in efficiency. Determining the best regularization coefficient in training objectives often requires multiple training runs, whereas our method can adaptively determine the optimal value for $\\alpha$ in a minute, as acknowledged in the Strengths section of Reviewer **3RHj**.
>
> [Meng et al.] Simpo: Simple preference optimization with a reference-free reward.
>
> [Park et al.] Disentangling length from quality in direct preference optimization.
>
> ---
>
> ***W4. Theoretical explanation of why repeated use of the implicit reward model can enhance performance.***
>
> We hypothesize that the performance gains of our approach arise from the iterative DPO process combined with on-policy sampling. The key intuition behind on-policy sampling is that it allows the LLM to continually generate preference pairs, reducing the likelihood of negative examples over time. This is in contrast to repeatedly optimizing over a fixed set of negative examples, which may already have a low likelihood and thus contribute minimally to further improvements.
>
> In Section 4.1 of the original paper (now moved to $\\textrm{\\color{blue}Appendix A}$ for better logical flow and readability), we provide a theoretical analysis highlighting the advantages of learning with on-policy samples. Furthermore, we empirically validate the effectiveness of the implicit reward in our response to ***W5***.
>
> ---
>
> ***W5. Advantages of implicit reward model compared with the external reward model.***
>
> As discussed in our response to ***W1***, this paper aims to answer **whether a DPO-tuned LLM can be further improved without relying on external models**, such as a stronger LLM or an external reward model. Therefore, comparisons with external reward models are outside the scope of this work.
>
> To ensure a fair comparison within our framework, we trained a scalar reward model using the same offline preference dataset employed for DPO training. We refer to this model as the **internal reward model (IntlRM)**. The IntlRM training follows the standard setting of OpenRLHF [Hu et al.], and we have included the training scripts in the updated Supplementary Material for reproducibility.
>
> We compared the IntlRM with the DPO implicit reward model in $\\textrm{\\color{blue}Table. 5}$ of the revised paper. The results indicate that the DPO implicit reward achieves higher accuracy than the IntlRM trained on the same dataset. Furthermore, it surpasses the performance of an external reward model, ERM-555k, trained on substantially more data. This implies that the implicit reward model offers advantages when evaluating its own generated data, though the scalar reward models in general excel on a wider range of tasks when the preference data is abundant, as demonstrated by ArmoRM-Llama3-8B-v0.1, which was trained on 1M preference data. Based on these findings, we argue that the implicit reward is a competitive option and its superiority can potentially lead to better performance in iterative training.
>
> [Hu et al.] OpenRLHF: An Easy-to-use, Scalable and High-performance RLHF Framework

---

> ### Author Response · Authors · 2024-11-25
> **Looking forward to further feedback**
>
> Dear Reviewer 6mfr,
>
> Thank you once again for your constructive feedback. We would like to kindly remind you that we have clarified our contributions, highlighted our method's significant performance improvements, and included comparisons with additional reward models in $\\textrm{\\color{blue}Section 5. 4}$ and $\\textrm{\\color{blue}Table. 5}$ of the paper revision.
>
> ---
>
> As the discussion period is coming to a close in two days, we look forward to your response and would be happy to address any further comments or questions you may have.
>
> Best,
>
> The Authors

---

### Comment · Reviewer_3RHj · 2024-11-12
**Further comment from reviewer 3RHj**

Reviewer 3RHj here – since my score is an outlier, I want to add a brief comment in support of the paper. While I agree with reviewer A3Uc that a comparison with the method of He et al. [1] would strengthen the paper, I think that the method presented remains a helpful contribution even without this comparison. This is because it does not involve any secondary models such as reward models, making it easier for practitioners to use. In my opinion, the experiments presented provide enough evidence for practitioners who are looking for simple extensions of DPO to be interested in trying this approach (modulo the main weakness mentioned in my review).

[1] Semi-Supervised Reward Modeling via Iterative Self-Training, https://arxiv.org/abs/2409.06903

---

### Author Response · Authors · 2024-11-20
**Summary of Paper Revision**

We thank all reviewers for their constructive feedback, and we have responded to each reviewer individually. We have also uploaded a **Paper Revision** including additional results and clarifications:

$\\textrm{\\color{blue}Section 5.4}$ (Page 9): Comparison of implicit reward model (ours) with scalar reward model (Reviewer **6mfr,A3Uc**).

$\\textrm{\\color{blue}Appendix B}$ (Page 14-15): Results of AlpacaEval 2 across two base models over three iterations (Reviewer **3Zep**).

$\\textrm{\\color{blue}Table. 7}$ (Page 16): Comparison with length-regularized DPO (Reviewer **3Zep**).

$\\textrm{\\color{blue}Appendix A}$ (Page 14): We have reorganized the paper by moving Section 4.1 to Appendix A. (Reviewer **A3Uc,3Zep**).

$\\textrm{\\color{blue}Table 8}$ (Page 17): Results of PPO with implicit rewards on AlpacaEval 2 (Reviewer **3RHj**).

$\\textrm{\\color{blue}Table 9 \\& 10}$ (Page 18): Hyperparameter tuning for $\\gamma$ (Reviewer **3RHj**).

---

### Meta-Review · Area_Chair_Nxrx · 2024-12-21

**Metareview:**

This paper provides a new approach for self-improvement in LLMs using implicit DPO rewards. The idea is to first perform DPO on a preference data, and then generate online samples, and use the learned implicit rewards to generate extra preference data and keep training. This idea makes sense and to my knowledge is novel.

Strengths:
1. Novel idea for self-improvement that avoids training a separate reward model or relying on it.
2. Results show that it does better than using LLM judge for selecting preference
3. Surprisingly, DICE does better than training a reward model on the preference data (IntlRM) and an external reward model.

Weakness:
1. Unclear how it would fair against a big space of preference learning literature, especially literature on reward learning. Authors do present useful baselines, particularly, using LLM judge and training a reward model.

Overall, I think the idea makes sense and the results are positive. I especially like that DICE does better than using LLM judge and doesn't require training in another reward model which is important from a practical point of view. Further, it is possible that implicit rewards are better for in-distribution tasks over general-purpose rewards. For this reason, I am recommending acceptance.

That said, I have a concern that as there are a lot of self-improvement learning methods using preferences and we don't quite have a grip on the best strategies given the early stages of this field. However, I think it is hard for a single paper outside the frontier labs to do an exhaustive evaluation of all proposed methods on a variety of tasks.

**Additional Comments On Reviewer Discussion:**

Reviewers discussed this paper thoroughly. The main concerns raised were

1. Why not use an external reward (reviewer A3Uc and 6mfr)? Authors and reviewer 3RHj argued that using implicit rewards avoids training a separate model. I like this argument and see its practical value. The authors also went ahead and compared with an approach that trains the reward model on the preference data (IntlRM) and showed that DICE work better.

2. Concerns about hyperparameters being chosen on the test set (reviewer 3RHj)? The authors did a clever study where they selected hyperparameters on one task and used it to test on another. This satisfies reviewer 3RHj who raised this concern.

3. Novelty of length bias and comparison with past length bias work (reviewer A3Uc and 6mfr): I think the authors addressed this concern well. The main advantage here is that since implicit rewards are used to construct preference data and the length penalty is added in the shaping term, it can be optimized using equation 6 which finds the smallest penalty such that the length of winning preference and losing preference are not too different. This makes sense.

I think the authors have done a decent job of addressing these concerns. I do hear concerns from reviewer A3Uc on evaluating the reward model approach fairly without spending enough time tailoring its hyperparameters. Neverthless, I feel this paper will generate some discussion in the field and will be tried by members of the community.

---

### Decision · Program_Chairs · 2025-01-22

Accept (Poster)